*Resource*

# Mouse models to investigate in situ cell fate decisions induced by p53

Elizabeth Lieschke [ID][1,2,7], Annabella F Thomas [ID][1,2], Andrew Kueh[1,2,3,4], Georgia K Atkin-Smith[1,2],
Pedro L Baldoni [ID][1,2], John E La Marca [ID][1,2,3,4], Savannah Young[1], Allan Shuai Huang [ID][1,2],
Aisling M Ross [ID][1,5], Lauren Whelan[1], Deeksha Kaloni[1,2], Lin Tai [ID][1,3], Gordon K Smyth [ID][1,6],
Marco J Herold [ID][1,2,3,4], Edwin D Hawkins[1,2], Andreas Strasser [ID][1,2,8 ✉] & Gemma L Kelly [ID][1,2,8 ✉]

## Abstract

**Investigating how transcription factors control complex cellular processes requires tools that enable responses to be visualised at the single-cell level and their cell fate to be followed over time. For example, the tumour suppressor p53 (also called TP53 in humans and TRP53 in mice) can initiate diverse cellular responses by transcriptional activation of its target genes: *Puma* to induce apoptotic cell death and *p21* to induce cell cycle arrest/cell senescence. However, it is not known how these processes are regulated and initiated in different cell types. Also, the context-dependent interaction partners and binding loci of p53 remain largely elusive. To be able to examine these questions, we here developed knock-in mice expressing triple-FLAG-tagged p53 to facilitate p53 pull-down and two p53 response reporter mice, knocking tdTomato and GFP into the *Puma/Bbc3* and *p21* gene loci, respectively. By crossing these reporter mice into a p53-deficient background, we show that the new reporters reliably inform on p53-dependent and p53-independent initiation of both apoptotic or cell cycle arrest/senescence programs, respectively, in vitro and in vivo.**

**Keywords** Apoptosis; Cell Cycle Arrest; Cancer; p53/TRP53/TP53; Reporter Mice
**Subject Categories** Autophagy & Cell Death; Cancer; Methods & Resources

## Introduction

The tumour suppressor TP53/TRP53 (also called p53) is a transcription factor that, in response to diverse stresses, such as oncogene activation, DNA damage or nutrient deprivation, activates an array of cellular responses to prevent tumour development. TP53/TRP53 binds as a homo-tetramer to a consensus motif that is present in the regulatory regions of ~500 direct target genes (el-Deiry et al, 1992; Funk et al, 1992; Riley et al, 2008). By controlling the expression of these target genes, TP53/TRP53 activates pathways to apoptotic cell death, cell cycle arrest/cell senescence, DNA damage repair and adaptation of cellular metabolism (Boutelle and Attardi, 2021; Kastenhuber and Lowe, 2017; Levine, 2020; Vousden and Lane, 2007). Amongst the proteins encoded by TP53/TRP53 direct target genes, the pro-apoptotic BH3-only proteins PUMA and (to a lesser extent NOXA) are essential to initiate TP53/TRP53-induced apoptosis (Jeffers et al, 2003; Villunger et al, 2003) and, conversely, the cyclin-dependent kinase inhibitor p21 is critical for TP53/TRP53-induced cell cycle arrest and cellular senescence (Deng et al, 1995).

It has not been resolved which cellular processes activated by TP53/TRP53 are critical for its tumour suppressive functions. It appears likely that the relative importance of the different TP53/TRP53-activated cellular responses for tumour suppression may vary between cell types undergoing neoplastic transformation and with respect to the nature of the oncogenic lesions that drive tumorigenesis (Kastenhuber and Lowe, 2017; Thomas et al, 2022). For example, defects in apoptosis caused by the loss of the pro-apoptotic TRP53 target gene *Bbc3* that encodes PUMA accelerate c-MYC-driven lymphomagenesis (Hemann et al, 2004; Michalak et al, 2009), but TRP53 can also prevent tumour development in the absence of its ability to induce apoptosis and cell cycle arrest/cell senescence (Brady et al, 2011; Li et al, 2012; Valente et al, 2013). Unbiased RNA interference-based functional screens in primary cells in mice revealed that coordination of DNA damage repair is critical for TRP53-mediated suppression of spontaneous tumour development (Janic et al, 2018). This agrees with the extensive genomic aberrations found in many mutant TP53-driven human cancers (Boutelle and Attardi, 2021; Kastenhuber and Lowe, 2017; Levine, 2020; Vousden and Lane, 2007).

Of note, it has not been resolved why upon activation of TP53/TRP53, some non-transformed cells, for example, lymphoid ones, will

[1]The Walter and Eliza Hall Institute (WEHI), Melbourne, VIC, Australia. [2]Department of Medical Biology, The University of Melbourne, Melbourne, VIC, Australia. [3]Olivia Newton-John Cancer Research Institute, Melbourne, VIC, Australia. [4]School of Cancer Medicine, La Trobe University, Melbourne, VIC, Australia. [5]School of Medicine, Bernal Institute, Limerick Digital Cancer Research Centre & Health Research Institute, University of Limerick, Limerick, Ireland. [6]School of Mathematics and Statistics, The University of Melbourne, Parkville, VIC, Australia. [7]Present address: Oncogene Biology Laboratory, Francis Crick Institute, London, United Kingdom. [8]These authors contributed equally: Andreas Strasser, Gemma L Kelly. ✉E-mail: strasser@wehi.edu.au; gkelly@wehi.edu.au

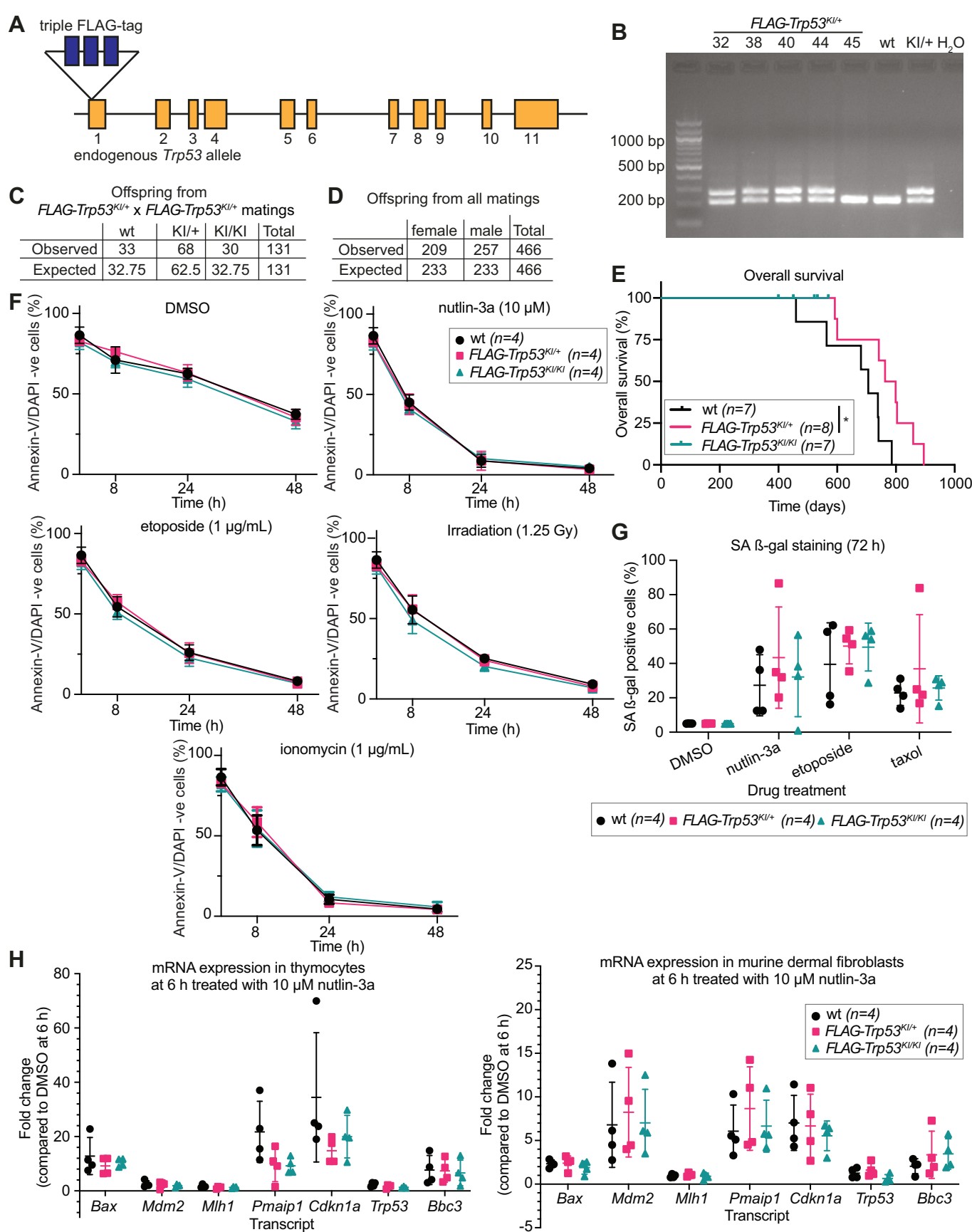

**Figure 1.** **The addition of an N-terminal triple-FLAG tag into the *Trp53* locus does not perturb the TRP53 function.**

(A) Schematic of the structure of the genetically modified *Trp53* allele. The coding sequence for the triple-FLAG tag was inserted after the start codon of the *Trp53* gene. (B) PCR analysis showing correct insertion of the coding sequences for the triple-FLAG tag into the genetically modified *Trp53* allele. A band of 225 bp indicates the presence of a wt (unmodified) allele. A band of 291 bp indicates a *FLAG-Trp53* allele. Each lane represents DNA taken from an individual mouse (#32, 38, 40, 44, 45) followed by DNA from a control wt mouse, control DNA from a *FLAG-Trp53$^{KI/+}$* mouse, and a water-only control. (C) Table with observed and expected genotype distribution from inter-crosses of heterozygous *FLAG-Trp53$^{KI/+}$* mice. (D) Table with observed and expected sex distribution from all matings that could give rise to *FLAG-Trp53$^{KI/+}$* or *FLAG-Trp53$^{KI/KI}$* mice. Frequencies compared by Chi-square test p = 0.0262. (E) Kaplan–Meier survival curve showing overall survival of wt, *FLAG-Trp53$^{KI/+}$*, and *FLAG-Trp53$^{KI/KI}$* mice. Differences in animal survival were compared using the Log-rank (Mantle–Cox) test, *p values ≤0.05. Mice were censored if harvested when healthy for use in experiments. (F) Thymocytes from wt, *FLAG-Trp53$^{KI/+}$* and *FLAG-Trp53$^{KI/KI}$* mice were treated for 48 h in vitro with DMSO (vehicle control), 10 μM nutlin-3a, 1 μg/mL etoposide, 1 μg/mL ionomycin or exposed to one dose of 1.25 Gy γ-radiation and then kept for 48 h in culture. Cell viability was measured at various time points by flow cytometry after staining with fluorochrome-conjugated annexin-V and DAPI. The percentages of live, i.e. annexin-V/DAPI double-negative cells, are plotted. p values were calculated using a two-way ANOVA using Dunnett's correction for multiple tests. Data were presented as mean ± SD. n = 4 mice of each genotype. (G) Mouse dermal fibroblasts (MDFs) were derived from wt, *FLAG-Trp53$^{KI/+}$* and *FLAG-Trp53$^{KI/KI}$* mice and treated for 72 h with DMSO (vehicle control), 10 μM nutlin-3a, 1 μg/mL etoposide or 250 nM taxol. Cellular senescence was measured as a percentage of fluorescein-di-β-ᴅ-galactopyranoside (FDG) positive cells by flow cytometry. p values were calculated using a two-way ANOVA using Dunnett's correction for multiple tests. Data were presented as mean ± SD. n = 4 mice of each genotype. (H) RNA was extracted from thymocytes and MDFs from wt, *FLAG-Trp53$^{KI/+}$* and *FLAG-Trp53$^{KI/KI}$* mice after they had been treated for 6 h with DMSO (vehicle control) or 10 μM nutlin-3a in vitro. qRT-PCR was performed to determine the mRNA levels of the TRP53 target genes *Bax*, *Mdm2*, *Mlh1*, *Pmaip1/Noxa*, *Cdkn1a/p21*, *Bbc3/Puma*, and the *Trp53* mRNA levels. Data were normalised using the ΔΔCT method, using *Hmbs* as a housekeeping gene. The data were plotted as fold-change of drug-treated compared to DMSO-treated cells. Data were presented as mean ± SD. n = 4 mice of each genotype. p values were calculated using a two-way ANOVA using Dunnett's correction for multiple tests. All statistical tests were non-significant (i.e. had a p value >0.05). Source data are available online for this figure.

undergo apoptosis (Clarke et al, 1993; Lowe et al, 1993; Strasser et al, 1994) whereas others, such as fibroblasts, do not die but instead undergo cell cycle arrest and cell senescence (Deng et al, 1995). It has been stated that cell fate is simply determined by *Puma* expression being induced by TP53/TRP53 only in those cells that will die but not in the cells that will survive and undergo cell cycle arrest/senescence (Boutelle and Attardi, 2021; Kastenhuber and Lowe, 2017; Levine, 2020; Vousden and Lane, 2007). This view has, however, been challenged by transcriptional analyses showing that TP53/TRP53 activation causes an increase in *Puma* and *p21* mRNA in both cell types that die and ones that survive and undergo cell cycle arrest (Brady et al, 2011; Moyer et al, 2020; Valente et al, 2016). Single-cell RNAseq analysis can identify which sets of genes are induced in an individual cell after TP53/TRP53 activation, but this technique does not allow follow-up analysis of the fate of such a cell because they need to be lysed to extract the RNA.

To advance research on the functions of TP53/TRP53, and on cell fate determination more generally, we have generated new mouse strains as powerful resources for future investigations. Knock-in of a triple-FLAG tag at the N-terminus of TRP53 did not affect its function but allowed for reliable CUT&RUN as well as proteomic analyses using antibodies against FLAG. Reporter mice with knock-in of GFP into the *Cdkn1a/p21* locus or knock-in of tdTomato into the *Bbc3/Puma* locus, respectively, allow for reliable detection of TRP53-dependent, as well as TRP53-independent, induction of these critical inducers of cell cycle arrest/cell senescence or apoptosis, respectively, in vitro and even in live animals. This illustrates the considerable potential of these new mouse resources, which will generally be made available for future exploration into TRP53 and other regulators of cell fate.

## Results

### N-terminally triple-FLAG tagged TRP53 mice

Although several high-quality TP53/TRP53 chromatin immuno-precipitation (ChIP) and CUT&RUN data sets have been published

(Delbridge et al, 2019; Hafner et al, 2020; Kenzelmann Broz et al, 2013; Li et al, 2020; Resnick-Silverman et al, 2023; Sanchez et al, 2014), the rabbit antiserum used in most of these studies is a non-renewable resource. Therefore, and since several laboratories, including ours, have found other commercially available antibodies against TP53/TRP53 not well suited for ChIP and CUT&RUN analyses, we developed a novel mouse resource that allows for reproducible pull-down of TRP53 proteins for this and other applications. To this end, we inserted a triple-FLAG tag at the N-terminus of TRP53 in its endogenous locus in mice on a C57BL/6 background (Fig. 1A). PCR analysis and DNA sequencing confirmed correct targeting of the *Trp53* gene locus (Figs. 1B and EV1A).

On a C57BL/6 background, around 60–75% of female *Trp53$^{−/−}$* mice die during embryogenesis due to defects in neural tube closure (Delbridge et al, 2019; Jacks et al, 1994), and 100% of *Trp53$^{−/−}$* mice (or mice with hypomorphic alleles of *Trp53*) develop cancer, mostly thymic T cell lymphoma, within 300 days (Donehower et al, 1992; Jacks et al, 1994; Valente et al, 2013). Notably, inter-crosses of heterozygous hereafter called *FLAG-Trp53$^{KI/+}$* mice yielded homozygous *FLAG-Trp53$^{KI/KI}$* as well as heterozygous *FLAG-Trp53$^{KI/+}$* mice and wt mice at the expected Mendelian ratio (1:2:1) (Fig. 1C) with a roughly 1:1 sex distribution (Fig. 1D; p value = 0.0262). Moreover, of 7 *FLAG-Trp53$^{KI/KI}$* homozygous and 9 *FLAG-Trp53$^{KI/+}$* heterozygous mice followed for up to 12-24 months, none developed cancer (Figs. 1E and EV1B; Appendix Table S1). Collectively, these findings demonstrate that the knock-in of the triple-FLAG tag at the N-terminus does not impair critical functions of TRP53 in embryogenesis or tumour suppression in *FLAG-Trp53$^{KI}$* mice.

Functional assays confirmed that cells from the *FLAG-Trp53$^{KI/KI}$* homozygous mice exhibited normal responses to stimuli that activate TRP53 or act in a TRP53-independent manner. Murine thymocytes spontaneously undergo apoptosis in vitro, which can be accelerated by TRP53-activating stimuli as well as stimuli that promote apoptosis through TRP53-independent processes (Strasser et al, 1991). Upon treatment with etoposide, γ-irradiation or the MDM2 inhibitor nutlin-3a (Vassilev et al, 2004) (all TRP53-dependent apoptotic stimuli) or

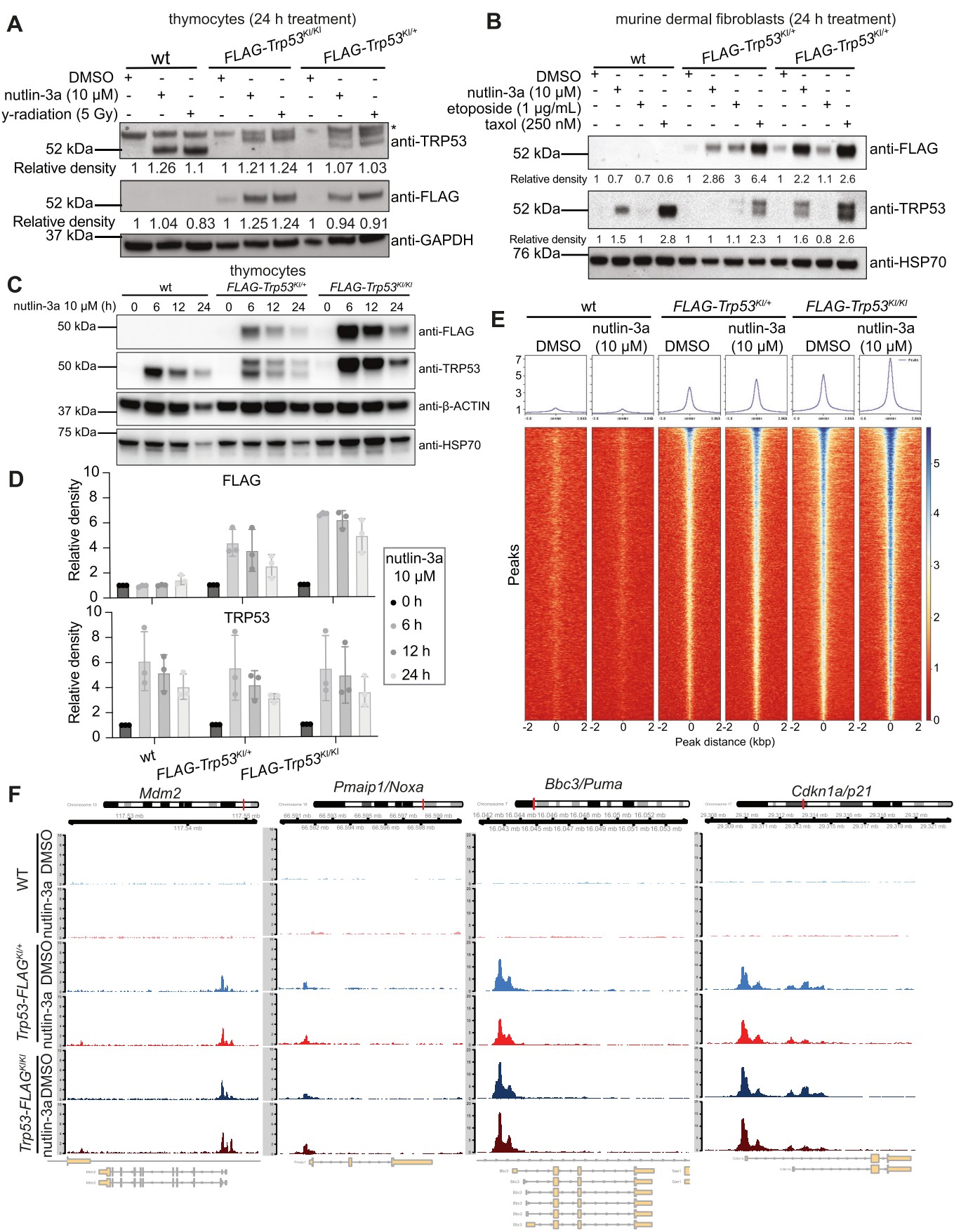

◄ **Figure 2. Antibodies against FLAG can be used to detect TRP53 in cells from the *FLAG-Trp53* mice.**

(A) Thymocytes from wt, *FLAG-Trp53^{KI/+}* and *FLAG-Trp53^{KI/KI}* mice were treated for 24 h with DMSO (vehicle control), 10 μM nutlin-3a or 5 Gy γ-radiation in the presence of the broad-spectrum caspase inhibitor QVD-oPH (to prevent degradation of protein due to apoptosis). Western blot analysis was performed on protein extracts, probing with antibodies against TRP53, FLAG and GAPDH (the latter used as a protein loading control). The asterisk indicates a non-specific band detected when probing for TRP53. Relative density for TRP53 and FLAG bands (normalised to GAPDH) are listed. (B) MDFs from wt, *FLAG-Trp53^{KI/+}* and *FLAG-Trp53^{KI/KI}* mice were treated for 24 h with DMSO (vehicle control), 10 μM nutlin-3a, 1 μg/mL etoposide or 250 nM taxol in the presence of QVD-oPH. Western blot analysis was performed on protein extracts, probing with antibodies against TRP53, FLAG and HSP70 (the latter used as a protein loading control). Relative density for TRP53 and FLAG bands (normalised to HSP70) are listed. (C) Thymocytes from wt, *FLAG-Trp53^{KI/+}* and *FLAG-Trp53^{KI/KI}* mice were treated with 10 μM nutlin-3a for 0, 6, 12 or 24 h in the presence of the broad-spectrum caspase inhibitor QVD-oPH. Western blot analysis was performed on protein extracts, probing with antibodies against TRP53, FLAG, HSP70 and β-ACTIN (the latter two used as protein loading controls). (D) Bar graph showing summary data of relative protein densities of TRP53, and TRP53-FLAG proteins quantitated from Western blots from three individual mice for each genotype normalised to β-ACTIN, and displayed as mean ± SD. (E) Heatmap from global analysis of CUT&RUN data showing the read coverage 2 kbp up- and downstream of enriched peaks found genome-wide in MDFs derived from wt, *FLAG-Trp53^{KI/+}* and *FLAG-Trp53^{KI/KI}* mice, treated with either DMSO (vehicle) or 10 μM nutlin-3a for 6 h. Peaks were called with MACS2. (F) FLAG antibody CUT&RUN tracks for the known TRP53 target genes *Mdm2*, *Pmaip1/Noxa*, *Bbc3/Puma* and *Cdkn1a/p21*. MDFs from wt, *FLAG-Trp53^{KI/+}* and *FLAG-Trp53^{KI/KI}* mice were treated in vitro for 6 h with either DMSO (vehicle control) or 10 μM nutlin-3a. Data shown are from MDFs from $n = 3$-4 mice of each genotype pooled in analysis, normalised to library size. Source data are available online for this figure.

ionomycin (TRP53-independent apoptotic stimulus) in culture, thymocytes from *FLAG-Trp53^{KI/KI}* and *FLAG-Trp53^{KI/+}* mice underwent apoptosis at the same rate as those from wt mice (Fig. 1F). Moreover, upon treatment with nutlin-3a, etoposide or taxol, cultured mouse dermal fibroblasts (MDFs) from *FLAG-Trp53^{KI/+}* and *FLAG-Trp53^{KI/KI}* mice entered cellular senescence-like MDFs from wt mice (Fig. 1G). Consistent with normal function of FLAG-TRP53 protein, substantial induction of classical TRP53 target genes, such as *Cdkn1a/p21*, *Puma/Bbc3*, *Bax*, *Mlh1*, *Mdm2*, *Pmaip1/Noxa* and *Trp53* itself was observed in thymocytes and MDFs from *FLAG-Trp53^{KI/+}* and *FLAG-Trp53^{KI/KI}* mice 6 h after treatment with nutlin-3a (Fig. 1H) or γ-radiation (Fig. EV1C).

Western blot analysis was performed on extracts from *FLAG-Trp53^{KI/KI}*, *FLAG-Trp53^{KI/+}* and wt thymocytes to detect the wt TRP53 and the FLAG-TRP53 proteins in cells. In untreated thymocytes the levels of both wt TRP53 and FLAG-TRP53 were very low but they both increased substantially, and importantly comparably, after treatment with nutlin-3a or γ-radiation (Fig. 2A). Antibodies against TRP53 detected both the wt TRP53 as well as the FLAG-TRP53 proteins, the latter exhibiting higher molecular weight, as expected (Fig. 2A). In contrast, antibodies against FLAG detected only the FLAG-TRP53 proteins in the thymocytes from the *FLAG-Trp53^{KI/KI}* and *FLAG-Trp53^{KI/+}* mice (Fig. 2A). A similar pattern was seen in MDFs from *FLAG-Trp53^{KI/+}* mice, *FLAG-Trp53^{KI/KI}* mice, and wt mice, treated with nutlin-3a, etoposide or taxol (Fig. 2B). We also confirmed that the addition of a triple-FLAG tag at the N-terminus did not affect the stability and degradation of the TRP53 protein (Fig. 2C,D). Overall, these data demonstrate that FLAG-TRP53 protein is expressed in unstressed, as well as stressed, cells at levels comparable to wt TRP53 protein.

Next, we tested whether FLAG-TRP53 pulldown can be used effectively for CUT&RUN analysis to identify the binding of TRP53 to its target genes. MDFs from *FLAG-Trp53^{KI/KI}*, *FLAG-Trp53^{KI/+}* and wt (negative control) mice were treated for 6 h in vitro with either nutlin-3a or DMSO (vehicle control). Cell extracts were made and subjected to pulldown using an antibody against FLAG. Global analysis of the peaks where there was enrichment of TRP53 binding demonstrated that there were basal levels of binding in DMSO-treated cells, as expected, and that binding to these loci was increased after treatment with nutlin-3a (Fig. 2E; Dataset EV1). DNA sequence analysis revealed reliable detection of binding of FLAG-TRP53 to known TRP53 binding sites in the genome, such as the regulatory regions of the *Puma/Bbc3* and *Cdkn1a/p21* genes,

in nutlin-3a treated cells from *FLAG-Trp53^{KI/KI}* and *FLAG-Trp53^{KI/+}* mice (Fig. 2F). No signal was detected when performing FLAG antibody pulldown on nutlin-3a treated MDFs from wt mice.

Collectively, these findings demonstrate that addition of a triple-FLAG tag to endogenous TRP53 did not perturb its expression or function and that cells from these mice can be used for reliable CUT&RUN analysis as well as proteomic studies to identify TRP53 binding sites in the genome or TRP53 interacting proteins, respectively.

### *p21-IRES-GFP* reporter mice

The cyclin-dependent kinase inhibitor p21, encoded by a direct TRP53 target gene, is essential for TRP53-induced cell cycle arrest at the G1/S boundary and entry into a senescent state (Deng et al, 1995). Thus, tracking the expression of *p21* mRNA is a reliable marker for the induction of these TRP53-activated cellular responses. We therefore generated *p21-IRES-GFP* reporter mice by inserting 3′ of the *p21* coding sequences into an internal ribosomal entry sequence (IRES) followed by the coding sequence for GFP (Fig. 3A). PCR analysis confirmed correct targeting of the *p21* locus (Fig. 3B).

First, we verified that the modification to the *p21* gene locus did not impact p21 expression and function. Upon treatment with nutlin-3a or etoposide, MDFs from *p21-IRES-GFP^{KI/KI}* and *p21-IRES-GFP^{KI/+}* mice ceased DNA synthesis and underwent cell cycle arrest at rates similar to wt MDFs (Figs. 3C and EV2A; Appendix Table S2). Moreover, thymocytes from *p21-IRES-GFP^{KI/+}* mice underwent apoptosis after treatment with these agents in a manner similar to wt thymocytes (Fig. 3D). Western Blot and qPCR analyses demonstrated that thymocytes and MDFs from *p21-IRES-GFP^{KI/+}* reporter mice had similar levels of both p21 protein and *p21* mRNA as the corresponding cells from wt mice (Fig. 3E,F). These findings demonstrate that the knock-in of the IRES and coding region for GFP did not perturb the expression and function of p21.

Next, we examined whether the *p21-IRES-GFP* reporter was detectable in cells. Analysis of MDFs from *p21-IRES-GFP^{KI/KI}* and *p21-IRES-GFP^{KI/+}* mice by flow cytometry revealed markedly higher GFP signal compared to wt MDFs at steady state (Fig. 4A). This is consistent with the observation that unstressed MDFs express basal levels of *p21* mRNA and p21 protein (Fig. EV2B,C). Upon treatment with nutlin-3a, etoposide or taxol, the levels of GFP

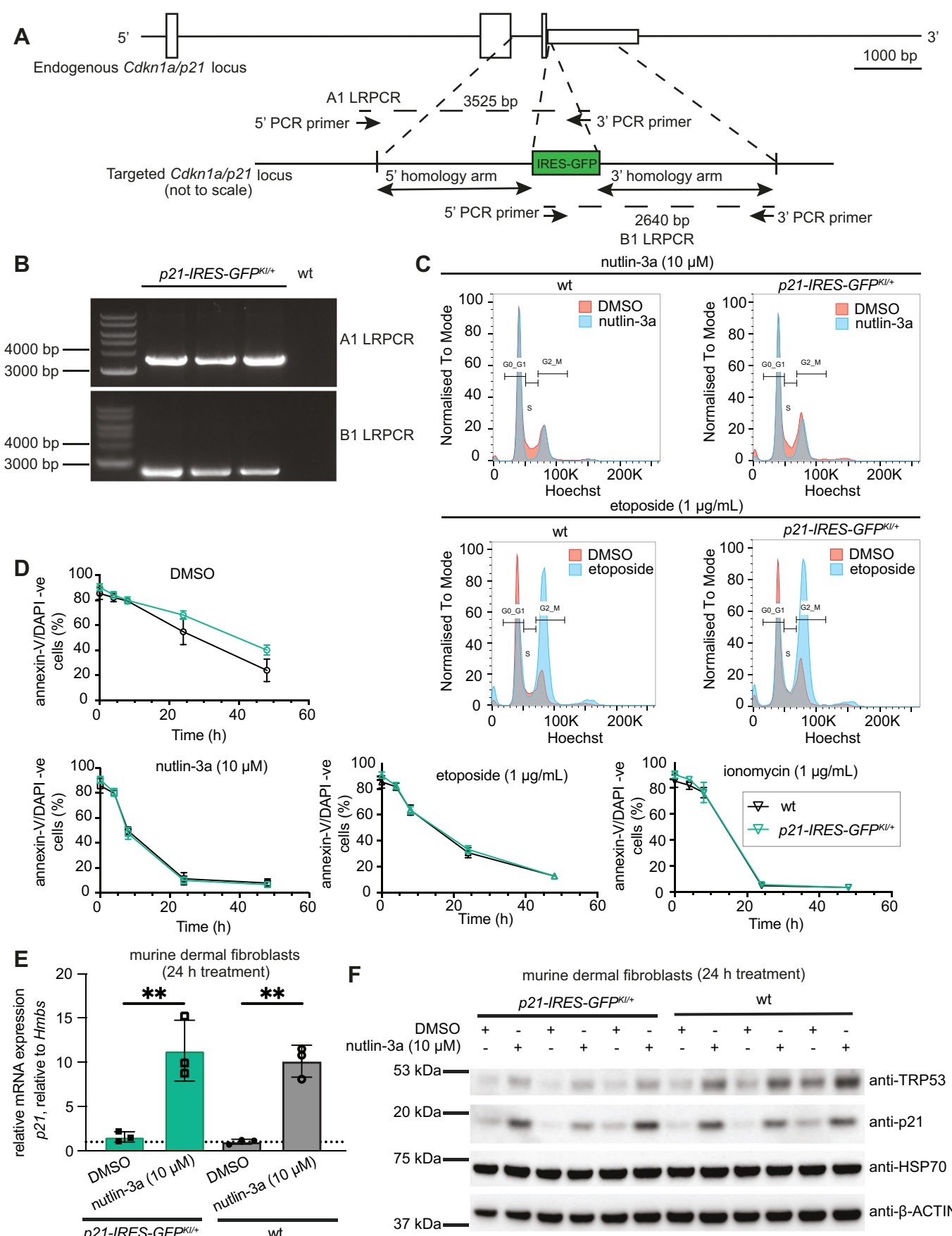

**Figure 3.  Generation and characterisation of *p21-IRES-GFP* reporter mice.**

(A) Schematic of the targeted *Cdkn1a/p21* locus to generate the *p21-IRES-GFP* reporter mice (not to scale), depicting where the long-range PCR primers that determine correct integration bind. (B) Agarose gel of DNA products from a long-range PCR to detect integration of the targeting construct. The first three lanes contain DNA from an individual *p21-IRES-GFP^{KI/+}* mouse, followed by DNA from a wt mouse in lane 4. All mice had the correct integration. (C) Representative histograms of the cell cycle distribution of MDFs from *p21-IRES-GFP^{KI/+}* or wt mice that had been treated for 24 h with either DMSO (vehicle control; in red), 10 μM nutlin-3a or 1 μg/mL etoposide (overlaid in blue). Cell cycle distribution was determined by staining with Hoechst 33342 followed by flow cytometric analysis at 24 h of drug treatment. Data were representative of $n = 6$–18 mice for each genotype from 5 independent experiments. (D) Thymocytes were extracted from *p21-IRES-GFP^{KI/+}* or wt mice, treated for 48 h in vitro with either DMSO (vehicle control), 10 μM nutlin-3a, 1 μg/mL etoposide or 1 μg/mL ionomycin. Cell survival was determined by flow cytometric analysis after staining with fluorochrome-conjugated Annexin-V plus DAPI. The percentages of live thymocytes (annexin-V/DAPI double negative) are graphed. p-values were calculated using a two-way ANOVA using Sidak's correction for multiple tests. No significant *p* values were found. Data were presented as mean ± SD. $n = 3$–6 mice per genotype. (E) qRT-PCR analysis of RNA extracted from MDFs from wt and *p21-IRES-GFP^{KI/+}* mice that had been treated for 24 h with either DMSO (vehicle control) or 10 μM nutlin-3a. Data were normalised using the ΔΔCT method, using *Hmbs* as a housekeeping gene. The data were plotted as a fold-change of drug-treated compared to the mean of the DMSO-treated cells. Data were presented as mean ± SD. $n = 3$ mice of each genotype. *p* values were calculated using a two-way ANOVA using Dunnett's correction for multiple tests. **p values ≤0.01. (F) Western blot analysis of proteins extracted from MDFs from wt and *p21-IRES-GFP^{KI/+}* mice that had been treated for 24 h with either DMSO (vehicle control) or 10 μM nutlin-3a. Probing for β-ACTIN was used as a loading control. Created with BioRender.com. Source data are available online for this figure.

increased substantially in the *p21-IRES-GFP^{KI/KI}* and *p21-IRES-GFP^{KI/+}* MDFs (Figs. 4B and EV2D–F). The increase in the *p21-IRES-GFP* reporter caused by treatment with nutlin-3a was entirely TRP53 dependent because this was not seen in *Trp53^{−/−};p21-IRES-GFP^{KI/+}* MDFs (Figs. 4B and EV2E,F). Interestingly, the basal *p21-IRES-GFP* reporter levels in unstressed MDFs were not markedly diminished by the absence of TRP53 (Fig. 4B). Slightly lower levels of p21 protein were detected by Western blotting in TRP53 deficient cells compared to wt cells (Fig. EV2B,C; Appendix Fig S1). This could be explained by the sensitivity of the antibody against p21 used or an effect of TRP53 on p21 protein synthesis or stability, in addition to its ability to activate transcription of the gene for p21. The levels of induction of the *p21-IRES-GFP* reporter after treatment with etoposide or taxol were diminished, but not abolished, by the absence of TRP53 (Figs. 4B and EV2E,F). This demonstrates that depending on the conditions, p21 expression can be regulated in a TRP53-dependent or TRP53-independent manner.

In thymocytes from *p21-IRES-GFP^{KI/KI}* and *p21-IRES-GFP^{KI/+}* mice, we did not detect the *p21-IRES-GFP* reporter either in steady state or after treatment with nutlin-3a, etoposide or ionomycin, despite them undergoing TRP53-mediated cell death (Fig. EV3A,B). We hypothesised that the low level of the *p21-IRES-GFP* reporter was due to these cells residing in the G₀ state. In mature T lymphocytes, which also reside in a G₀ state, we could also not detect *p21-IRES-GFP* reporter activity, even after treatment with nutlin-3a or other cytotoxic agents (Fig. 4C). However, after mitogenic stimulation, the *p21-IRES-GFP* reporter became readily detectable in proliferating T lymphocytes and its levels increased further after treatment with nutlin-3a, taxol or etoposide (Figs. 4D and EV3C,D). Proliferating T lymphocytes from *p21-IRES-GFP^{KI/KI}* mice had approximately double the GFP signal compared to proliferating T lymphocytes from *p21-IRES-GFP^{KI/+}* mice. However, the fold-induction between cells from the two genotypes was similar (Fig. EV3E). The absence of TRP53 in mitogen-activated cells from *Trp53^{−/−};p21-IRES-GFP^{KI/+}* mice abrogated the nutlin-3a (and other TRP53-dependent stimuli) induced increase in the levels of the reporter but it did not diminish the basal levels of this reporter in the absence of stress (Figs. 4D and EV3F). To determine if the increase in the reporter was transient, we treated MDFs with nutlin-3a, taxol, etoposide or abemaciclib for 24 h before removing the medium containing the drug and replacing it with a fresh drug-free medium. From this experiment, we could see that following

the initial increase in the levels of GFP in cells treated with nutlin-3a, peaking at 48 h, the reporter levels subsequently started to decrease (Figs. 4E and EV3G).

The ultimate utility of reporter mice is to visualise the induction of defined cellular responses in situ in live mice. We, therefore, tested whether the *p21-IRES-GFP* reporter could be detected in cells in vivo. Nutlin-3a cannot be easily administered to mice. We, therefore, subjected *p21-IRES-GFP^{KI/+}* and control wt mice to 5 Gy γ-radiation, as a means of activating TRP53 (Lotem and Sachs, 1993; Strasser et al, 1994) and performed intra-vital microscopy of the bone marrow (Hawkins et al, 2016) (Appendix Fig. S2A,B). This approach is advantageous as it can provide high-resolution spatial and temporal information. Moreover, mice can be recovered after individual imaging sessions and re-imaged over a series of days to allow the tracking of in vivo cellular dynamics. Basal levels of the *p21-IRES-GFP* reporter were observed in cells throughout the bone marrow of untreated mice (Fig. 5A,B). At 48 h after whole-body γ-irradiation, increases in the levels of GFP were seen in cells in the bone marrow of *p21-IRES-GFP^{KI/+}* mice but not in wt mice (Fig. 5A,B; Appendix Fig. S2C). Little increase in the activity of this reporter was seen in γ-irradiated *Trp53^{−/−};p21-IRES-GFP^{KI/+}* mice (Fig. 5A,B), confirming that the induction was TRP53-dependent. The levels of the *p21-IRES-GFP* reporter in untreated mice were not reduced by the absence of TRP53 (Fig. 5A,B), demonstrating that its basal expression is independent of TRP53.

To investigate expression of the *p21-GFP* reporter beyond the bone marrow, we performed multiphoton imaging on excised organs from *p21-IRES-GFP^{KI/+}* mice, and wt controls, with and without prior whole-body γ-irradiation (Fig. 5C–Z; Appendix Fig. S3). We detected increased GFP expression after γ-irradiation in lymph nodes, spleen and the kidney but not in the heart or liver.

Collectively these findings demonstrate that the *p21-IRES-GFP* reporter informs reliably on the expression of p21 and thus p21-initiated cellular processes in response to both TRP53-dependent as well as TRP53-independent stimuli in vitro and even within the whole mouse.

### *Puma-tdTomato* reporter mice

The pro-apoptotic BH3-only member of the BCL-2 family PUMA/BBC3, encoded by a direct TRP53 target gene (Han et al, 2001; Nakano and Vousden, 2001; Yu et al, 2001), is critical for TRP53

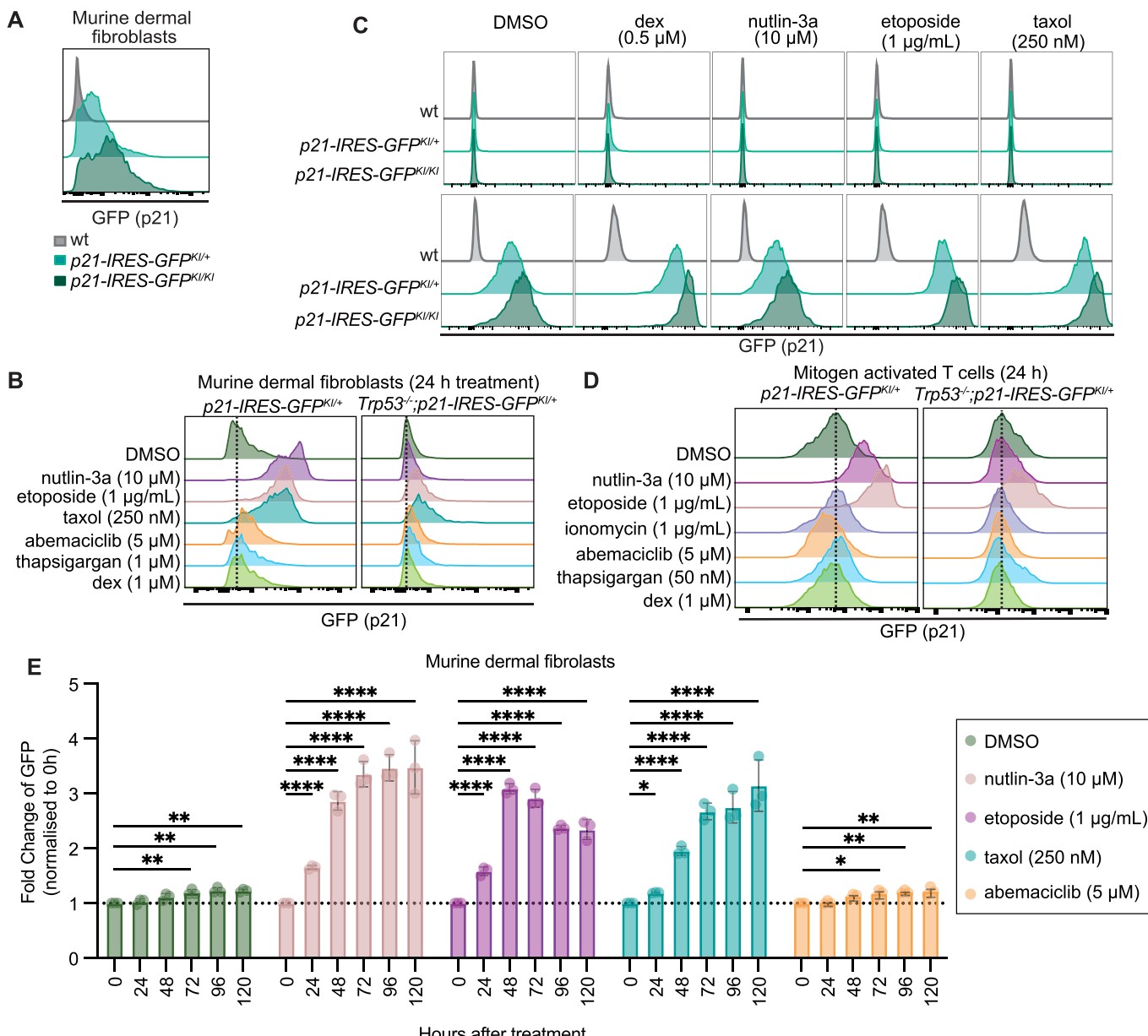

**Figure 4. GFP expression increases in cells from *p21-IRES-GFP* reporter mice after treatment with cytotoxic agents that act in either a TRP53-dependent or a TRP53-independent manner.**

(A) Flow cytometry histograms showing the levels of GFP in untreated MDFs from a *p21-IRES-GFP^{KI/+}* and a *p21-IRES-GFP^{KI/KI}* reporter mouse and, as a control, a wt mouse. (B) Flow cytometry histograms showing the levels of GFP in MDFs from a *p21-IRES-GFP^{KI/+}* mouse and a *Trp53^{-/-};p21-IRES-GFP^{KI/+}* mouse that had been treated for 24 h with DMSO (vehicle control), 10 μM nutlin-3a, 1 μg/mL etoposide, 250 nM taxol, 5 μM abemaciclib, 1 μM thapsigargin or 1 μM dexamethasone. Representative histograms from *n* = 3–7 independent cell cultures per genotype are shown. (C) Flow cytometry histograms showing the levels of GFP in resting (top row) or mitogen-activated (bottom row) T cells from a *p21-IRES-GFP^{KI/+}* and a *p21-IRES-GFP^{KI/KI}* reporter mouse and, as a control, a wt mouse. T cells were isolated from lymphoid organs with no stimulation or stimulated with antibodies against CD3 and CD28 for 48 h before being treated with DMSO (vehicle control), 0.5 μM dexamethasone, 10 μM nutlin-3a, 1 μg/mL etoposide or 1 μg/mL ionomycin for 24 h. Representative histograms from *n* = 4 independent cell cultures per genotype are shown. (D) Flow cytometry histograms comparing the levels of GFP in mitogen-activated T cells from a *p21-IRES-GFP^{KI/+}* mouse and a *Trp53^{-/-};p21-IRES-GFP^{KI/+}* mouse after treatment with DMSO (vehicle control), 10 μM nutlin-3a, 1 μg/mL etoposide, 1 μg/mL ionomycin, 5 μM abemaciclib, 50 nM thapsigargin or 1 μM dexamethasone for 24 h. Representative histograms from *n* = 4 independent cell cultures per genotype are shown. (E) Summary plots of GFP expression in murine dermal fibroblasts displayed as fold-change relative to level at 0 h. Cells were treated with DMSO (vehicle control), 10 μM nutlin-3a, 1 μg/mL etoposide, 250 nM taxol or 5 μM abemaciclib for 24 h before drug-containing medium was removed and replaced with non-drug-containing medium. *p* values were calculated using a linear model using Sidak's correction for multiple tests. Data displayed as mean ± SD from *n* = 3 cultures derived from independent mice of each genotype. *p* value * ≤0.05, ** ≤0.01 and **** ≤0.0001. Source data are available online for this figure.

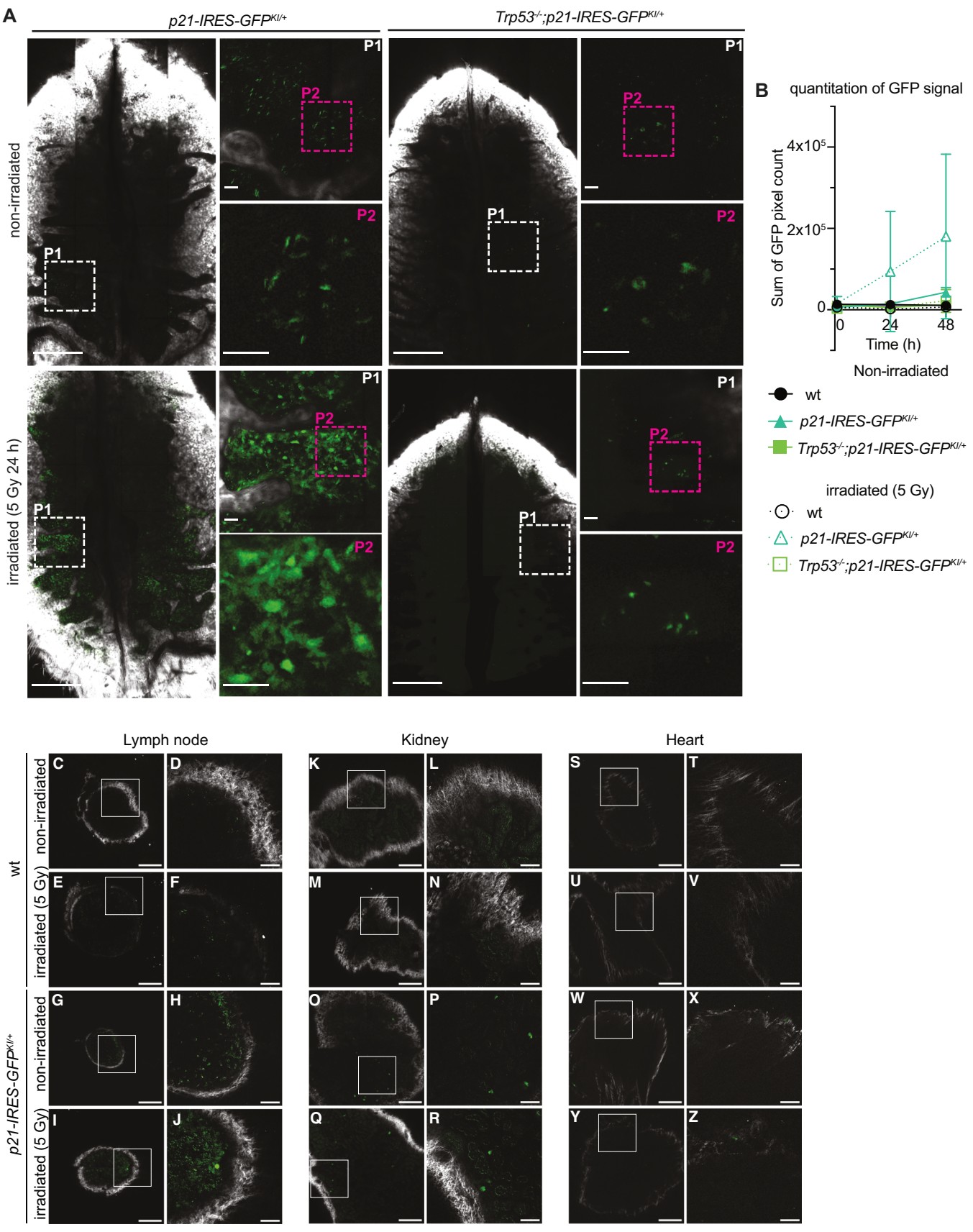

**Figure 5. GFP expression can be detected by intra-vital microscopy in *p21-IRES-GFP* reporter mice after γ-irradiation.**

(A) Representative multiphoton microscopy images of the calvarium of a *p21-IRES-GFP*$^{KI/+}$ mouse and a *Trp53*$^{-/-}$;*p21-IRES-GFP*$^{KI/+}$ mouse 24 h after administration of a single dose of 5 Gy γ-irradiation to activate TRP53. Data shown are representative of $n = 3$ mice for each genotype from three independent experiments. Scale bars are 500 μm for the whole calvarium image and 50 μm in P1 and P2. (B) Quantification of the GFP pixel count from images described in A (including wt mice as controls). Data were quantified from an $n = 3$–4 mice for each genotype and treatment and displayed as mean ± SD. (C–Z) Representative multiphoton microscopy images of the lymph node, kidney and heart from a wt and a *p21-IRES-GFP*$^{KI/+}$ mouse 18 h after administration of a single dose of 5 Gy whole-body γ-irradiation. Tissues from non-irradiated mice were used as a control. Scale bars for C, E, G, I, K, M, O, Q, S, U, W and Y = 200 μm. Scale bars for D, F, H, J, L, N, P, R, T, V, X and Z = 50 μm. Images from a *p21-IRES-GFP*$^{KI/}$ mouse are also displayed in Appendix Fig. S3.

induced apoptotic cell death (Jeffers et al, 2003; Villunger et al, 2003). Thus, tracking the expression of *Puma* mRNA is a reliable marker for the induction of the TRP53-activated pathway to apoptosis. We, therefore, generated *Puma-tdTomato* reporter mice by replacing the coding sequence for *Puma* with that for tdTomato (Fig. 6A). PCR analysis confirmed the correct targeting of the *Puma* locus (Fig. 6B).

First, we examined the impact of the modification to the *Puma* locus on the behaviour of cells. Loss of one allele of *Puma* causes a minor, albeit significant, reduction in TRP53-induced apoptosis in lymphoid cells, whereas the complete absence of PUMA causes resistance (Michalak et al, 2008; Villunger et al, 2003). Since the coding sequences for PUMA were removed and replaced with those for tdTomato, we expected cells from heterozygous *Puma-tdTomato*$^{KI/+}$ reporter mice to show similar minor resistance to TRP53-induced apoptosis as those from *Puma*$^{+/-}$ mice and those from *Puma-tdTomato*$^{KI/KI}$ homozygous mice to be markedly resistant, akin to those from *Puma*$^{-/-}$ mice. Indeed, this is what was found when thymocytes were treated with nutlin-3a or etoposide (Fig. 6C). As expected, thymocytes from *Trp53*$^{-/-}$;*Puma-tdTomato*$^{KI/+}$ mice were resistant to nutlin-3a and etoposide (Fig. EV4A). To not impact experiments by reduced apoptosis, we only used *Puma-tdTomato* mice in a heterozygote state for future experiments.

After treatment with nutlin-3a MDFs from *Puma-tdTomato*$^{KI/+}$ mice ceased DNA synthesis, underwent cell cycle arrest and entered senescence at rates similar to wt MDFs (Figs. 6D and EV4B; Appendix Table S3). As expected, MDFs from *Trp53*$^{-/-}$;*Puma-tdTomato*$^{KI/+}$ mice did not undergo cell cycle arrest after treatment with nutlin-3a (Fig. EV4B,C; Appendix Table S3). qPCR and Western Blot analyses of nutlin-3a treated or vehicle-treated thymocytes and MDFs from *Puma-tdTomato*$^{KI/+}$ reporter mice showed slightly lower levels of *Puma* mRNA and PUMA protein, respectively, as seen in the corresponding cells from *Puma*$^{+/-}$ mice (Figs. 6E,F and EV4D,E). However, there was still an increase in expression upon nutlin-3a treatment, indicating the cells from the *Puma-tdTomato*$^{KI/+}$ reporter mice were still responsive to the treatment. These findings demonstrate that the knock-in of the coding region for tdTomato had the expected minor impact on PUMA expression and PUMA-induced apoptosis.

Next, we examined the expression of the *Puma-tdTomato* reporter by flow cytometry. As expected, untreated MDFs from *Puma-tdTomato*$^{KI/+}$ mice displayed markedly higher tdTomato signal than MDFs from non-reporter *Puma*$^{+/-}$ mice (Fig. 7A), consistent with the observation that unstressed MDFs express basal levels of *Puma* mRNA and PUMA protein (Han et al, 2001). Upon treatment with nutlin-3a, etoposide or taxol, the levels of tdTomato

increased markedly in the *Puma-tdTomato*$^{KI/+}$ MDFs (Figs. 7B and EV5A,B). The increase in the *Puma-tdTomato* reporter caused by treatment with nutlin-3a was entirely TRP53 dependent because this was not seen in *Trp53*$^{-/-}$;*Puma-tdTomato*$^{KI/+}$ MDFs (Figs. 7B and EV5A,B). The increases in reporter levels after treatment with etoposide or taxol were diminished but not abrogated by the absence of TRP53 (Fig. 7B). Interestingly, the basal expression of tdTomato in unstressed MDFs was not diminished by the absence of TRP53 (Fig. 7B).

Consistent with observations that *Puma* mRNA and PUMA protein can be detected in wt thymocytes (Michalak et al, 2008), untreated thymocytes from *Puma-tdTomato*$^{KI/+}$ mice displayed increased tdTomato signal compared to thymocytes from non-reporter *Puma*$^{+/-}$ mice (Fig. 7C). After treatment with nutlin-3a, etoposide or various other cytotoxic agents, the levels of the *Puma-tdTomato* reporter increased substantially in thymocytes from *Puma-tdTomato*$^{KI/+}$ mice (Figs. 7D and EV5C,D). The absence of TRP53 in cells from *Trp53*$^{-/-}$;*Puma-tdTomato*$^{KI/+}$ mice prevented the nutlin-3a induced increase in the levels of this reporter, yet reporter expression could still be induced in these cells by treatment with dexamethasone or PMA (Figs. 7D and EV5C,D), agents that induce apoptosis in a PUMA-dependent but TRP53-independent manner (Villunger et al, 2003). To determine if the *Puma-tdTomato* reporter expression was reversible, we treated MDFs for 24 h with nutlin-3a, taxol, etoposide or abemaciclib before removing the drug-containing medium and replacing it with fresh drug-free medium (Figs. 7E and EV5E). *Puma-tdTomato* reporter expression increased steadily during treatment with nutlin-3a until 48 h (i.e. for another 24 h after removal of the agent) and then plateaued. For treatment with etoposide, the levels of the *Puma-tdTomato* reporter continued to increase even after the drug was removed. Reporter expression in cells treated with taxol peaked at 48–72 h (i.e. for another 24–48 h after removal of the drug) and subsequently diminished (Figs. 7E and EV5E). Treatment with abemaciclib had only a minor impact on the levels of the *Puma-tdTomato* reporter throughout the time course of the experiment (Figs. 7E and EV5E).

We extended the analysis of the *Puma-tdTomato* mice to visualise the responses of cells in living mice to stress. We subjected *Puma-tdTomato*$^{KI/+}$ and control wt mice to 5 Gy γ-radiation and performed intra-vital microscopy of the bone marrow. Basal levels of the *Puma-tdTomato* reporter were observed throughout cells in the bone marrow of untreated mice (Fig. 8A,B). At 24 and 48 h after whole-body γ-irradiation, marked increases in the levels of the reporter were seen in bone marrow cells of the *Puma-tdTomato* mice but not the wt controls (Fig. 8A,B; Appendix Fig. S2D). Little increase in the levels of the reporter was seen in γ-irradiated

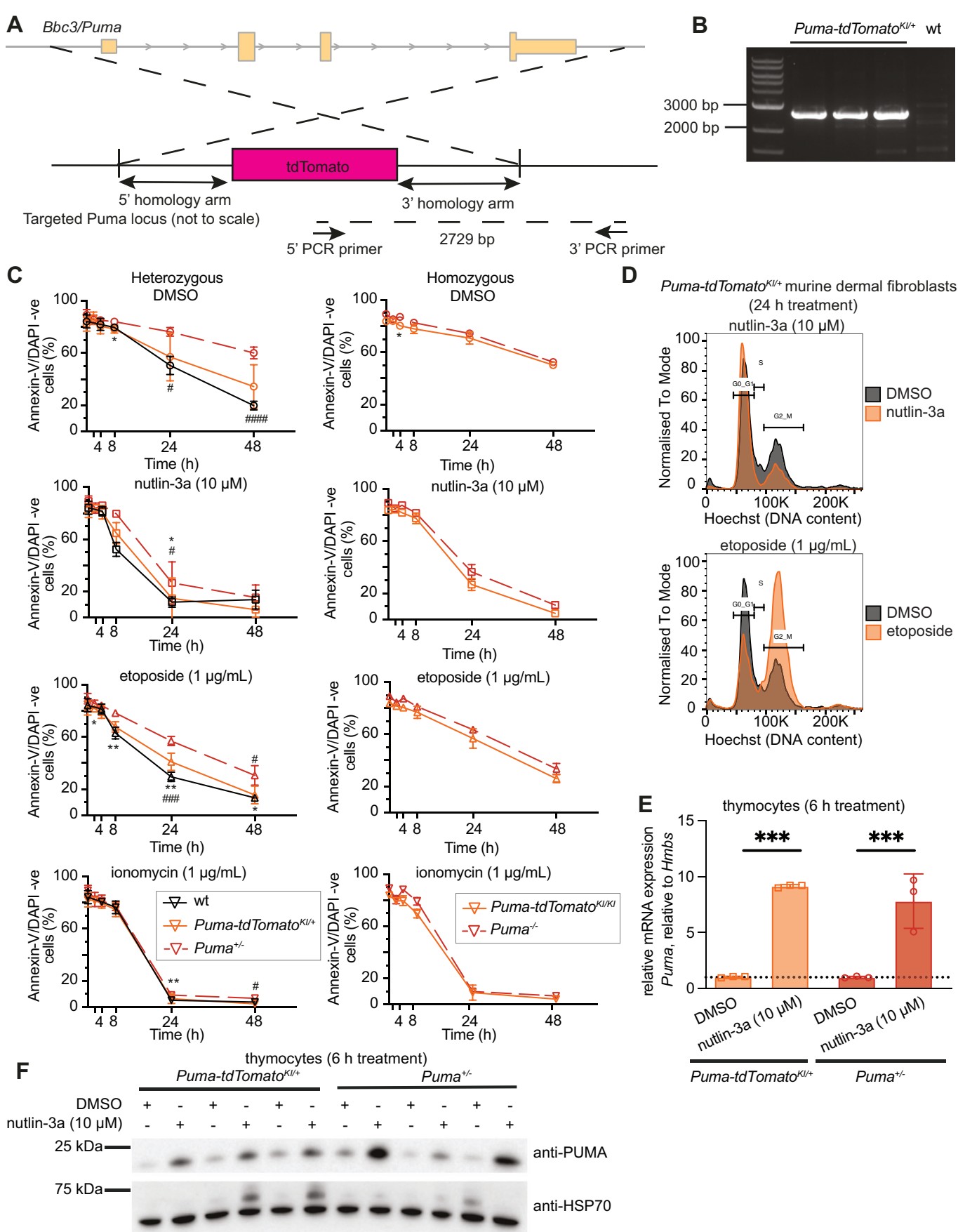

**Figure 6. Generation and characterisation of *Puma-tdTomato* reporter mice.**

(A) Schematic of the targeted *Bbc3/Puma* locus to generate the *Puma-tdTomato* reporter mice (not to scale), depicting where the long-range PCR primers that determine correct integration bind. (B) Agarose gel of DNA products from a long-range PCR to detect integration of the targeting construct. The first three lanes contain DNA from an individual *Puma-tdTomato*^KI/+^ mouse, followed by DNA from a wt mouse in lane 4. All mice had the correct integration. (C) Thymocytes were extracted from wt, *Puma*^+/−^, *Puma-tdTomato*^KI/+^, *Puma*^−/−^ and *Puma-tdTomato*^KI/KI^ mice, treated for 48 h in vitro with either DMSO (vehicle control), 10 μM nutlin-3a, 1 μg/mL etoposide or 1 μg/mL ionomycin. The percentages of live thymocytes (annexin-V/DAPI double negative) as determined by flow cytometric analysis are graphed. $p$ values were calculated using a two-way ANOVA using Sidak's correction for multiple tests. $n = 3$–6 mice per genotype. Data were presented as mean ± SD. * describe comparisons between wt and *Puma*^+/−^ mice, or between *Puma-tdTomato*^KI/KI^ and *Puma*^−/−^ mice. # symbols describe comparisons between *Puma* and *Puma-tdTomato*^KI/+^ mice. $p$ values = * or # ≤0.05, ** ≤0.01, ### ≤0.001, #### ≤0.0001. $n = 3$–5, mean ± SD. (D) Representative histograms for cell cycle distribution of MDFs from *Puma-tdTomato*^KI/+^ mice that had been treated for 24 h with DMSO (vehicle control; in black), overlaid with plots from cells treated for 24 h with 10 μM nutlin-3a or 1 μg/mL etoposide (in orange). Cell cycle distribution was determined by staining with Hoechst 33342 followed by flow cytometric analysis. Data were representative of $n = 3$ independent mice per genotype from three independent experiments. (E) qRT-PCR analysis of RNA extracted from thymocytes from *Puma*^+/−^ and *Puma-tdTomato*^KI/+^ mice that had been treated for 6 h with either DMSO (vehicle control) or 10 μM nutlin-3a in the presence of the broad-spectrum caspase inhibitor QVD-oPH. Data were normalised using the ΔΔCT method, using *Hmbs* as a housekeeping gene. Data were plotted as fold-change of drug-treated compared to the mean of the DMSO-treated cells. Data were presented as mean ± SD. $n = 3$ mice of each genotype. $p$ values were calculated using a two-way ANOVA using Dunnett's correction for multiple tests. $p$ values *** ≤0.001. (F) Western blot analysis of proteins extracted from thymocytes from *Puma*^+/−^ and *Puma-tdTomato*^KI/+^ mice that had been treated for 6 h with either DMSO (vehicle control) or 10 μM nutlin-3a in the presence of the broad-spectrum caspase inhibitor QVD-oPH. Probing for HSP70 was used as a loading control. Created with BioRender.com. Source data are available online for this figure.

*Trp53*^−/−^;*Puma-tdTomato*^KI/+^ mice (Fig. 8A,B), demonstrating that this induction is TRP53 dependent. The levels of the *Puma-tdTomato* reporter in untreated mice were not reduced by the absence of TRP53 (Fig. 8A,B), demonstrating that the basal expression of *Puma* is independent of TRP53.

To investigate the expression of the *Puma-tdTomato* reporter beyond the bone marrow, we performed multiphoton imaging on organs excised from *Puma-tdTomato*^KI/+^ mice and wt control mice, with and without prior whole-body γ-irradiation (Fig. 8C–Z; Appendix Fig. S3). *Puma-tdTomato* reporter expression was detectable at baseline, and increased upon γ-irradiation in the spleen, lymph nodes, kidneys and heart, but not the liver (Fig. 8C–Z; Appendix Fig. S3).

Collectively these findings demonstrate that the *Puma-tdTomato* reporter informs reliably on the expression of *Puma* mRNA and thus on the PUMA-initiated apoptotic programme activated by both TRP53-dependent as well as TRP53-independent stimuli in culture and in situ in live mice.

## Discussion

Well-designed technological advances and scientific tools are critical for answering fundamental biological questions in the most informative way. Here we describe three novel genetically engineered mouse strains that can provide insight into the function of the key tumour suppressor TP53/TRP53 and many other regulators of cell fate.

Genetically engineered mouse models provide renewable sources of robust materials for studies without relying on reagents (e.g. antibodies) specific to the protein of interest. Variability in the effectiveness of reagents is potentially one reason for the differences in lists of direct TP53/TRP53 target genes reported across different studies (Fischer, 2017; Fischer et al, 2016). Here, we sought to create a robust and reliable system to identify TRP53 target genes and proteins interacting with TRP53 by knocking in a triple-FLAG tag to the N-terminus of endogenous TRP53. The addition of this tag did not disrupt TRP53 function and could be used reliably to detect where TRP53 binds to DNA elements in the genome. In addition to the Western blot studies shown here, we believe cells

from *FLAG-Trp53* mice will be valuable resources for proteomic studies for identifying and confirming proteins that bind to TRP53 by using immunoprecipitation or mass spectrometry. This could be of particular interest to identify proteins that bind mutant TRP53 proteins that may be involved in exerting its reported dominant-negative or gain-of-function activities. This could be achieved by crossing the *FLAG-Trp53* mice with transgenic mice expressing an oncogene, such as *c-Myc*, in which mutations in *Trp53* are prevalent cooperating oncogenic lesions driving tumorigenesis, selecting those lymphomas that have sustained a mutation of interest in the *FLAG-Trp53* allele.

The two reporter mouse strains described here read out on two of the major cellular processes that TRP53 activates to suppress tumour development, namely apoptotic cell death and cell cycle arrest/cell senescence (Boutelle and Attardi, 2021; Thomas et al, 2022). Importantly, the insertion of the reporter constructs did not abrogate the functions of p21 or PUMA (for the latter in a heterozygous state only). In the *Puma-tdTomato* mice, we identified minor differences in the levels of *Puma* mRNA and PUMA protein expressed, but as we still observed induction of the mRNA and protein, this functionally had little impact. This may be because the cells have adapted to lower levels of this pro-apoptotic protein. To prevent the loss of the function of one allele of *Puma* that we see in our *Puma-tdTomato* reporter mice, we are currently generating *Puma-IRES-mCherry* mice in which the function of the targeted *Puma* allele should not be impaired.

Importantly, our new reporter mice cannot just be used for studies looking at TRP53 function. We have shown that we can detect induction of these reporters also after TRP53-independent stimuli, such as induction of the *Puma-tdTomato* reporter after exposure of lymphoid cells to PMA or dexamethasone. These two cytotoxic agents induce apoptosis in lymphoid cells in a manner that is in part dependent on PUMA but independent of TRP53 (Villunger et al, 2003). This means we can use the reporter mice to investigate how p21 and PUMA are regulated in a variety of settings including in cancer models. Of note, we can use cells, such as immortalised cell lines or tumour-derived cell lines, from these mice to perform CRISPR screens to search for positive and negative regulators of the expression of the *Cdkn1a/p21* and *Bbc3/Puma* genes and the pathways in which they function.

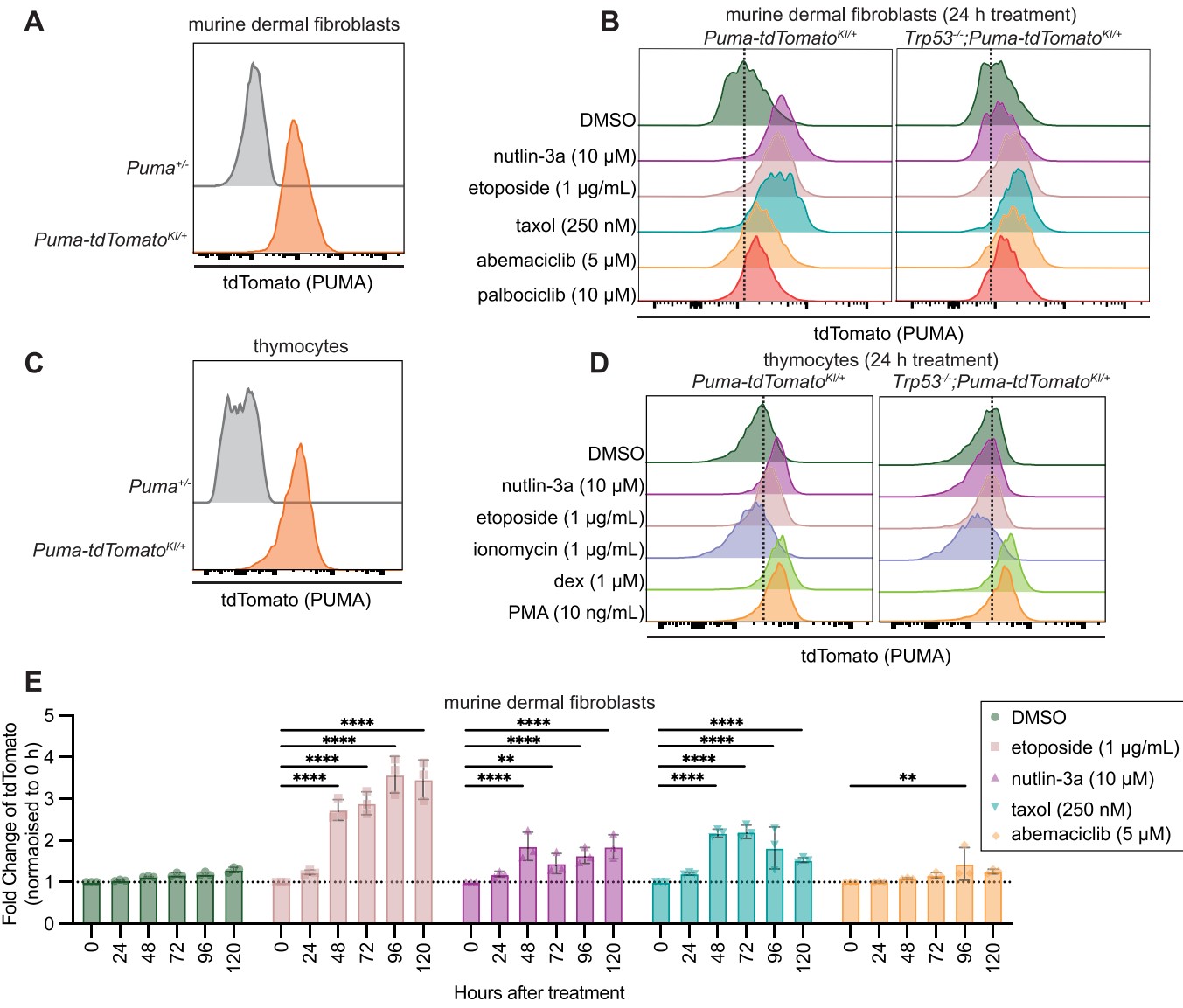

**Figure 7. tdTomato expression increases in cells from *Puma-tdTomato* reporter mice after treatment with cytotoxic agents that kill in either a TRP53-dependent or a TRP53-independent manner.**

(A) Flow cytometry histograms showing the levels of tdTomato in untreated *Puma-tdTomato*$^{KI/+}$ MDFs compared to untreated MDFs from a *Puma*$^{+/-}$ mouse. Data shown were representative of $n = 3$–5 for each genotype. (B) Flow cytometry histograms showing the levels of tdTomato in MDFs from a *Puma-tdTomato*$^{KI/+}$ mouse and MDFs from a *Trp53*$^{-/-}$; *Puma-tdTomato*$^{KI/+}$ mouse that had been treated for 24 h with DMSO (vehicle control), 10 μM nutlin-3a, 1 μg/mL etoposide, 250 nM taxol, 5 μM abemaciclib or 10 μM palbociclib. Representative histograms from $n = 5$ independent cell cultures for cells of each genotype are shown. (C) Flow cytometry histograms showing the levels of tdTomato in untreated *Puma-tdTomato*$^{KI/+}$ thymocytes compared to untreated thymocytes from a *Puma*$^{+/-}$ mouse. Data shown were representative of $n = 3$–5 mice for each genotype. (D) Flow cytometry histograms showing the levels of tdTomato in thymocytes from a *Puma-tdTomato*$^{KI/+}$ mouse and a *Trp53*$^{-/-}$; *Puma-tdTomato*$^{KI/+}$ mouse that had been treated for 24 h with DMSO (vehicle control), 10 μM nutlin-3a, 1 μg/mL etoposide, 1 μg/mL ionomycin, 1 μM dexamethasone or 10 ng/mL PMA in the presence of the broad-spectrum caspase inhibitor QVD-oPH. Representative histograms from $n = 3$ independent cell cultures for each genotype and treatment are shown. (E) Summary plots of tdTomato expression in murine dermal fibroblasts displayed as fold-change relative to the level at 0 h. Cells were treated with DMSO (vehicle control), 10 μM nutlin-3a, 1 μg/mL etoposide, 250 nM taxol or 5 μM abemaciclib for 24 h before drug-containing medium was removed and replaced with non-drug-containing medium. $p$ values were calculated using a linear model using Sidak's correction for multiple tests. Data displayed as mean ± SD from $n = 3$ cultures derived from independent mice of each genotype. $p$ value ** ≤0.01, **** ≤0.0001. Source data are available online for this figure.

We have conclusively demonstrated our reporter mice can be combined with intra-vital imaging approaches to visualise expression in situ. Thus, these models are powerful tools to study apoptosis and cell cycle control in vivo in as many different cell types as this technology allows. We have demonstrated tissue-specific activation of the two reporters. The *Puma-tdTomato* and *p21-IRES-GFP* reporter data shown here largely agree with published RNAseq analysis of tissues from γ-irradiated mice

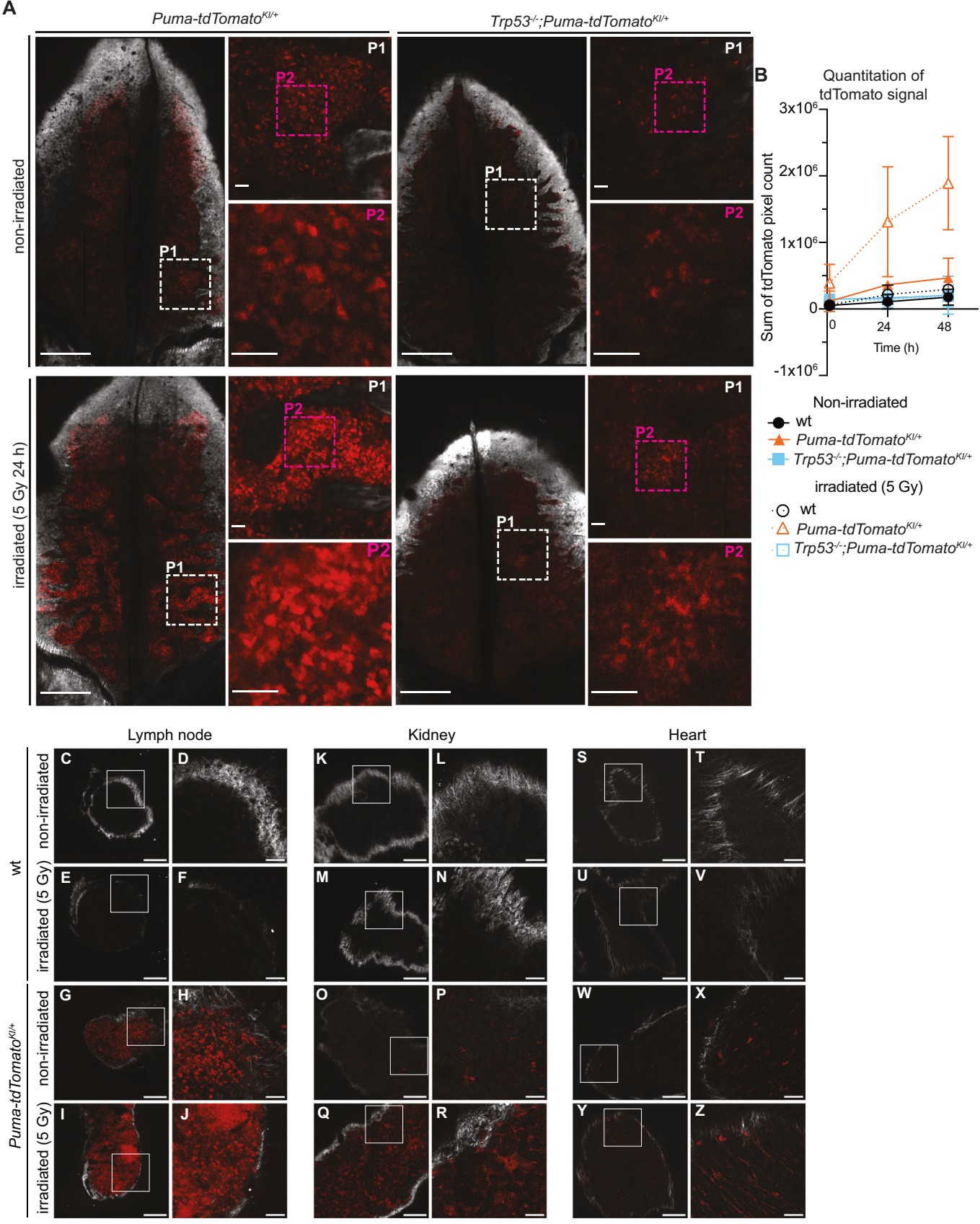

**Figure 8. tdTomato expression can be detected by intra-vital microscopy in *Puma-tdTomato* reporter mice after γ-irradiation.**

(A) Representative multiphoton microscopy images of the calvarium of a *Puma-tdTomato^{KI/+}* mouse and a *Trp53^{−/−};Puma-tdTomato^{KI/+}* mouse 24 h after administration of a single dose of 5 Gy γ-irradiation to activate TRP53. Data shown were representative of n = 3 independent experiments per mouse genotype and treatment. Scale bars are 500 μm for the whole calvarium image and 50 μm in P1 and P2. (B) Quantification of the tdTomato pixel counts from images described in A (including wt mice as controls). Data quantified from an n = 3–5 mice for each genotype and treatment and displayed as mean ± SD. (C) Representative multiphoton microscopy images of lymph nodes, kidney and heart from a wt and a *Puma-tdTomato^{KI/+}* mouse 18 h after administration of a single dose of 5 Gy whole-body γ-irradiation. Tissues from non-irradiated mice were used a control. Scale bars for C, E, G, I, K, M, O, Q, S, U, W and Y = 200 μm. Scale bars for D, F, H, J, L, N, P, R, T, V, X and Z = 50 μm. Images from a *Puma-tdTomato^{KI/+}* mouse are also displayed in Appendix Fig. S3.

(Resnick-Silverman et al, 2023); the only discrepancy found was for *p21* expression in the liver. This could be due to differences in experimental conditions, such as the time point for tissue analysis. Future experiments could be used to perform powerful fundamental studies in the context of whole animals with regards to toxicities of drug treatments, the emergence of cells exhibiting resistance to therapeutic intervention and basic developmental biology processes that underpin life.

Previous mice generated to report on *Cdkn1a/p21* expression have used lacZ or luciferase as the reporters (McMahon et al, 2016; Tinkum et al, 2011; Vasey et al, 2008). Notably, these reporters are both not suitable for in vivo fluorescent imaging. Additionally, the *Cdkn1a^{SUPER}* mouse strain (Torgovnick et al, 2018) was inadequate to answer our questions, as its cells over-express p21. Fluorescent reporters for both *Puma* and *p21* have previously been described (Goh et al, 2012). However, these reporters were not expressed from the endogenous loci, but instead from the *Hprt* locus, although containing *Trp53* binding sites in their 5′ UTR. The absence of other regulatory regions in the promoters and the lack of the overall genomic context makes these animals less suitable than our mice for investigations of TRP53-dependent expression of gene regulation and entirely unsuited to examine the TRP53-independent control of these genes.

An interesting finding in our study was that quiescent cells did not express GFP reporting on *p21* expression even after treatment with nutlin-3a to activate TRP53 or other cytotoxic agents. This is in contrast to mRNA expression data, which show that the levels of *p21* increase after TRP53 activation, such as in response to treatment with nutlin-3a (Valente et al, 2016). One possible explanation could be that the levels of *p21* mRNA in nutlin-3a treated thymocytes, even though increased compared to untreated thymocytes, are still not sufficiently high to give rise to substantial GFP reporter (and endogenous p21 protein) expression. Of note, thymocytes and quiescent mature T and B cells do not need to induce p21 expression to undergo cell cycle arrest, given that they reside in the $G_0$ state. Alternatively, our finding could be one example of many (de Sousa Abreu et al, 2009; Vogel and Marcotte, 2012) where protein and mRNA expression do not correlate, and this may deserve further investigation.

Using our new mouse models in vivo and in various cell types in vitro, we can now investigate the timing of when the initiators of cell cycle arrest/senescence or apoptotic cell death are first induced and their dynamics over the lifetime of a cell. We can also investigate if TRP53 has different interactions with DNA or other proteins in cells that have different levels of *p21* or *Puma* by generating mice that express both reporters and also FLAG-TRP53.

In conclusion, our new mouse resources will greatly enrich research in cancer biology, cell biology developmental biology, immunology and many other areas providing insight into TRP53 and also other regulators that initiate cell cycle arrest via induction of *p21* and/or cell death via induction of *Puma* to instruct cell fate.

# Methods

**Reagents and tools table**

| Reagent/resource | Reference or source | Identifier or catalogue number |
|---|---|---|
| **Experimental models** | | |
| Trp53 KO (M. musculus) | Tyler Jacks | B6.129S2-*Trp53^{tm1Tyj}* |
| FLAG-Trp53 (M. musculus) | This study | |
| p21-IRES-GFP (M. musculus) | This study | |
| Puma-tdTomato (M. musculus) | This study | |
| **Recombinant DNA** | | |
| N/A | | |
| **Antibodies** | | |
| Rabbit-anti-Trp53 (polyclonal) | Novacastra/Lecia | Cat # P53-PROTEIN-CM5 |
| Mouse-anti-FLAG (monoclonal, M2 affinity purified) | Sigma | Cat # F1804 |
| rabbit anit-GAPDH (monoclonal, 14C10) | Cell signaling technologies | Cat # 2118 |
| Mouse anti-HSP70 (monoclonal, N6) | In house | In House |
| Goat anti -Rabbit IgG-HRP secondary (polyclonal) | Southern Biotech | Cat # 4010-05 |
| Goat anti -Mouse IgG-HRP secondary (polyclonal) | Southern Biotech | Cat # 1010-05 |
| Annexin-V-FITC | Biolegend | #640905 |
| Anti-CD28 (monoclonal, 37.51) | In house | In House |
| Anti-CD3 (monoclonal, 2C11) | In house | In House |
| **Oligonucleotides and sequence-based reagents** | | |
| PCR primers - Trp53 KO genotyping | | 5′- TTATGAGCCACCCG AGGT 5′- TATACTCAGAGCCG GCCT 5′- TCCTCGTGCTTTACG GTATC |
| PCR primers - FLAG-Trp53 genotyping | This study | 5′ - CTCGAGGCTGATATCC GACTG 5′ – CCGGCTTCTGTCCTCC ATG |
| PCR primers - p21-IRES-GFP genotyping | This study | 5′ – AAAGTTCTCCCTGGCC CAAG 5′ – TTGTAATCGGGGATGT CGGC 5′ – CGGGCTAGACCCTCT ACGG |

| Reagent/resource | Reference or source | Identifier or catalogue number |
|---|---|---|
| *PCR primers - Puma-tdtomato genotyping* | This study | 5′ – TGGCCTGACATCTGT TCTCTC<br>5′ - ACTGCTTCCTTCACG ACATTC<br>5′ – GCAGGCAGCGTATAT ACAGG |
| **Chemicals, enzymes and other reagents** | | |
| Nutlin-3a | Cayman Chemicals | #18585 |
| etoposide | Ebewe Interpharma | AUST R 96641 |
| Dexamethasone | Sigma-Aldrich | D1756 |
| PMA | Sigma-Aldrich | P1585 |
| Thapsigargin | Sigma-Aldrich | T9033 |
| ionomycin | Sigma-Aldrich | I9657 |
| Encequidar | MedChem Express | HY-13646 |
| Hoechst 33342 | Sigma-Aldrich | #14533 |
| paclitaxel | Bristol Myers Squibb | |
| palbociclib | MedChemExpress | HY-50767 |
| abemaciclib | MedChemExpress | HY-16297A |
| DAPI | Sigma-Aldrich | 28718-90-3 |
| PI | Thermo Fisher Scientific | P1304MP |
| COMPLETE protease inhibitor cocktail tablets | Roche | #11697498001 |
| Dulbecco's Modified Eagle Medium (DMEM) | Gibco | Thermo Fisher #12491015 |
| RPMI-1640 | Gibco | Thermo Fisher #12633012 |
| ESC-qualified foetal bovine serum (FBS) | Gibco | Thermo Fisher #A4736301 |
| Asparagine | Sigma-Aldrich | #70-47-3 |
| GlutaMAX™-I Supplement | Gibco | Thermo Fisher #35050061 |
| MEM Non-Essential Amino Acids Solution | Gibco | Thermo Fisher #11140050 |
| 2-mercaptoethanol | Sigma-Aldrich | #M-6250 |
| Penicillin/Streptomycin | Gibco | Thermo Fisher #15140122 |
| IL7 | In house | N/A |
| IL2 | In house | N/A |
| **Software** | | |
| Gviz package | Hahne and Ivanek, 2016 | http://bioconductor.org/biocLite.R |
| ImageLab | Bio-Rad | https://www.bio-rad.com |
| Prism | GraphPad | https://www.graphpad.com |
| FlowJo™ v10 | BD Biosciences | https://www.bdbiosciences.com |
| CaseViewer | 3DHISTECH | https://www.3dhistech.com |
| Adobe Photoshop | Adobe | https://www.adobe.com |
| ImageJ2 (v2.9.0) | FIJI | https://imagej.net/software/fiji/downloads |

| Reagent/resource | Reference or source | Identifier or catalogue number |
|---|---|---|
| **Other** | | |
| Epicypher CUTANA ChIC/CUT&RUN kit v2 | Epicypher | Cat #14-1048 |
| SimpleChIP/ChIP-seq Multiplex Oligos for Illumina (Dual Index Primers | Cell Signalling Technology | Cat #47538 |
| NEBNext Ultra II DNA Library Prep Kit for Illumina | New England Biolabs | Cat #E7645L |

## Mouse strains

All animal experiments were conducted with the approval of the WEHI Animal Ethics Committee, following the guidelines set out by the Melbourne Directorate Animal Ethics Committee. $Trp53^{-/-}$ mice have been previously described (Jacks et al, 1994). They were originally generated on a mixed C57BL/6x129SV background but had been backcrossed with C57BL/6 mice for >20 generations prior to the commencement of the studies described here.

Several new mouse strains were developed for this research. These include *Trp53-N-terminal-triple-FLAG-Tag* knock-in mice, referred to as *FLAG-Trp53* mice as well as *Puma-tdTomato* and *p21-IRES-GFP* reporter mice. All new mouse strains were generated on a C57BL/6 background by the MAGEC lab at WEHI using CRISPR/Cas9 gene editing technology. *FLAG-Trp53* mice were generated by targeting and inserting vector DNA encoding a triple-FLAG tag into the endogenous *Trp53* locus. *Puma-tdTomato* reporter mice were generated by replacing the coding sequences for *Bbc3* (encodes PUMA) with the coding sequence for tdTomato. *P21-IRES-GFP* mice were generated by inserting an internal ribosome entry site (IRES) and the GFP coding sequence into the endogenous *p21* locus. Mice with correctly inserted DNA sequences, as determined by next-generation sequencing, were maintained on a C57BL/6 background.

## Genotyping of mice by PCR

Crude DNA was extracted from either a tailpiece or an ear clip obtained at weaning, using the DirectPCR Lysis buffer (Viagen #102-T) with 0.32 U/mL proteinase K (Sigma-Aldrich #P4850) at 56 °C for a minimum of 4 h until completion of tissue digestion. Proteinase K was inactivated at 86 °C for 20 min. PCR was performed using GoTaq Green Master Mix (Promega #M7123). An aliquot of 1 μL of DNA was added to 19 μL of master mix with primers. Primer sequences are as follows: $Trp53^{-/-}$ 5′-TTAT-GAGCCACCCGAGGT 5′-TATACTCAGAGCCGGCCT 5′-TCC TCGTGCTTTACGGTATC, *Puma-tdTomato* 5′–AAAGTTCTCCC TGGCCCAAG 5′-TTGTAATCGGGGGATGTCGGC 5′–CGGGCTA GACCCTCTACGG, *FLAG-Trp53* 5′-CTCGAGGCTGATATCCGA CTG 5′–CCGGCTTCTGTCCTCCATG, *p21-IRES-GFP* 5′–TGGCC TGACATCTGTTCTCTC 5′-ACTGCTTCCTTCACGACATTC 5′–GCAGGCAGCGTATATACAGG. Target sequences were ampli-fied using the following cycling conditions: $Trp53^{-/-}$ and *p21-IRES-GFP*—initial denaturation at 94 °C for 4 min followed by 30 cycles of 94 °C for 30 s, 55 °C for 30 s, 72 °C for 1 min, followed by a final extension at 72 °C for 5 min. *Puma-tdTomato*—initial denaturation at 94 °C for 4 min followed by 30 cycles of 94 °C for 40 s, 62 °C for

30 s, 72 °C for 1 min, followed by a final extension at 72 °C for 5 min. *FLAG-Trp53*—initial denaturation at 95 °C for 2 min followed by 35 cycles of 95 °C for 30 s, 60 °C for 30 s, 72 °C for 30 s, followed by a final extension at 72 °C for 5 min. PCR products were run on a 2% agarose gel in 1X TAE buffer (Sigma-Aldrich #T9650 diluted with water) containing 0.5 µg/mL ethidium bromide at 100 V for 40 min and visualised using the ChemiDoc XRS+ in the ImageLab programme.

## Preparation of single-cell suspensions from mouse tissues

Single-cell suspensions of thymocytes, spleen cells and lymph node cells were generated from the primary tissues of mice. The organs were harvested and placed in 2 mL BSS (150 mM NaCl, 3.7 mM KCl, 2.5 mM CaCl$_2$.2H$_2$O, 1.2 mM MgSO$_4$.7H$_2$O, 14.8 mM HEPES, prepared by the WEHI media kitchen). Thymi, spleens and lymph nodes were gently passed through a 100 µm mesh to generate a single-cell suspension. For spleen samples and some thymi, if there were many red blood cells, those were depleted by treatment with Red Cell Removal Buffer (prepared by the WEHI Media Kitchen) for 5 min at room temperature before washing twice with DPBS with 2% Heat Inactivated Foetal Bovine Serum (HI-FBS, Sigma, Batch #16K598 or #20K207). Cells were then counted and plated for further assays.

## Isolation of primary T cells

Naïve T cells were isolated from the spleen and lymph nodes of mice. Organs were harvested and gently passed through a 100 µm mesh to make a single-cell suspension. Cells were incubated with a biotinylated antibody cocktail (MAC-1 (MI/70) /CD19 (ID3) /CD44 (1M7.81) /B220 (RA3-6B2) /TER119 (Ter119) /NK1.1 (PK136)) that binds all unwanted cells, washed, and incubated with the MagniSort Streptavidin Negative Selection Beads (Invitrogen #MSNB-6002). Samples were placed on a magnet, and the supernatant was collected. Enrichment and purity of cells were assessed by flow cytometry after staining with antibodies against CD3 (17A2 or 145-2C11, produced in-house), CD4 (GK1.5, produced in-house), CD8 (53.6.7, BD Biosciences #563234), CD44 (1M7.81, produced in-house) and CD62L (MEL14 L-selectin, produced in-house) before plating for downstream assays. Primary naïve T cells were cultured in RPMI-1640 medium supplemented with GlutaMAX (Gibco #35050061), 10 mM HEPES (Gibco #15630-080), 1 mM sodium pyruvate (Gibco #11360-070), 1x non-essential amino acids (Sigma #M7145), 50 µM β-mercaptoethanol (Sigma-Aldrich #M-6250), 100 U/mL penicillin, 100 mg/mL streptomycin (Gibco #15140122) and 10% HI-FBS. Cells were cultured at 37 °C in 10% CO$_2$.

## Generation of murine dermal fibroblasts

Murine dermal fibroblasts (MDFs) were generated from the tails of 8–12 weeks-old mice as described previously (Lieschke et al, 2022). Bone was removed from the skin and then the skin was digested in 2.1 U/mL dispase II (Sigma-Aldrich #D4693) in DMEM (Gibco #11885-092) either at 37 °C for 1 h or at 4 °C overnight. The dermis was peeled from the epidermis and placed in 0.42 mg/mL Collagenase IV (Sigma-Aldrich #C5138) in DMEM at 37 °C for

1 h. The dermis was passed through a 100 µm mesh, and collagenase was washed away with DMEM. Cells were plated in two wells of a six-well plate in DMEM (Gibco #11885-092) supplemented with 100 U/mL penicillin, 100 mg/mL streptomycin (Sigma #P0781) and 10% HI-FBS. Cells were grown at 37 °C in 10% CO$_2$ and passaged every 2–3 days until natural cell senescence, usually 3 weeks after the start of the culture. All assays, unless specified otherwise, were performed on early passage (passage 2–5) fibroblasts.

## Western blotting

Cell lysates were made by resuspending cells in RIPA buffer (50 mM Tris.HCl pH 8, 150 mM NaCl, 1% NP-40, 0.5% sodium deoxycholate, 0.1% SDS) with cOmplete protease inhibitor (Roche #11697498001) for 25 min on ice, followed by a 10 min centrifugation at 13,000 rpm to clear cellular debris. Protein concentration was determined by the Bradford assay using the Bio-Rad Protein Assay Dye (#5000006), or by using the Pierce™ BCA protein kit (Thermo Fisher Scientific, #23225). About 10–25 µg of protein was denatured by incubating at 100 °C for 5 min with Laemmli buffer (0.25 M Tris.HCl pH 6.8, 40% glycerol, 0.8% SDS, 0.1% bromophenol blue, 10% β-mercaptoethanol). Samples were run on either a 10% or 4–12% Bis-Tris NuPage gel (Life Technologies) to separate proteins by size in MES buffer. Proteins were then transferred onto a nitrocellulose membrane using the iBlot 2 system (Thermo Fisher Scientific #IB21001, 20 V 1 min, 23 V 4 min, 25 V 2 min). Membranes were blocked in 5% skim milk in PBS-T (DPBS with 0.1% Tween-20). Membranes were then incubated with the primary antibodies listed in Table 1 diluted in 5% BSA in PBS-T. Membranes were washed in PBS-T before incubating in HRP-conjugated secondary antibodies listed in Table 1 specific to the relevant species of the primary antibody used. Luminata Forte Western HRP substrate (Millipore #WBLUF0500) was applied to the membrane shortly before imaging using the ChemiDoc XRS+ machine. Images were analysed using ImageLab software.

## RNA extraction, cDNA conversion and qRT-PCR analysis

Cells were collected directly into TRIzol Lysis Reagent (Invitrogen #15596026) and stored at −80 °C until RNA was extracted. RNA was extracted using the Phenol/chloroform method. The concentration and purity of RNA were determined using a DeNovix DS-11 Spectrophotometer/Fluorometer. RNA was determined as good quality if the 260:280 ratio was above 2.

Up to 1 µg of RNA was converted to cDNA using superscript III (Invitrogen #18080-051) and oligo-Dt primers following the manufacturer's protocol. qRT-PCR was performed using TaqMan probes in TaqMan™ Universal PCR Master Mix (Applied Biosystems #4304437). The list of primers used can be found in Table 2. Samples were run on Viia7 using Quantstudio software and analysed using the comparative threshold (ΔΔCT) method.

## CUT&RUN assay

About 500,000 MDFs were plated and left to settle for 24 h in an incubator with 10% CO$_2$ at 37 °C. The medium was replaced with medium containing nutlin-3a (10 µM, Cayman Chemicals #18585)

**Table 1.  Antibodies used for Western blot analysis.**

| Specificity | Origin | | Clone | Source/Company | Catalogue # |
|---|---|---|---|---|---|
| TRP53 | Rabbit | pAb | CM5 | Novocastra/Leica | P53-PROTEIN-CM5 |
| FLAG | Mouse | mAb | M2 (affinity purified) | Sigma-Aldrich | F1804 |
| GAPDH | Rabbit | mAb | 14C10 | Cell Signaling Technologies | 2118 |
| HSP70 | Mouse | mAb | N6 | In House | N/A |
| β-ACTIN | Mouse | mAb | AC-74 | Sigma-Aldrich | A2228 |
| P21 | Rabbit | pAb | N/A | Cell Signaling Technologies | 64016S |
| PUMA | Rabbit | mAb | E2P7G | Cell Signaling Technologies | 98672 |
| Rabbit IgG-HRP secondary | Goat | pAb | N/A | Southern BioTech | 4010-05 |
| Mouse IgG-HRP secondary | Goat | pAb | N/A | Southern BioTech | 1010-05 |

*pAb* polyclonal antibody, *mAb* monoclonal antibody.

or DMSO (vehicle control) and cells were incubated for 6 h. Adherent cells were dissociated with trypsin (0.25 mg/mL, Sigma #T4174), keeping both the supernatant and the trypsinised fractions for analysis. CUT&RUN was performed using the CUTANA ChIC/CUT&RUN kit v2 (Epicypher, 14-1408) following the manufacturer's protocol. Briefly, nuclei were isolated prior to the start of the assay. Nuclei were bound to activated conA beads. Samples were incubated with 0.5 μg of anti-FLAG antibody (Sigma, F1804) overnight. DNA was cut using pAG-MNase, and unbound fragments were washed away. Bound chromatin was released from the beads, and DNA was purified. Total recovered DNA was taken into library preparation using the NEBNext Ultra II DNA Library Prep Kit for Illumina (#E7645L). Indexing was performed using the SimpleChIP/ChIP-seq Multiplex Oligos for Illumina (Dual Index Primers) (#47538). $2 \times 75$bp paired-end read sequencing was performed on a NextSeq, and 10 million reads per sample was collected.

## CUT&RUN data analysis

Paired-end CUT&RUN sequence reads were aligned to the mouse reference genome mm39 with the *Rsubread* package and the *align* function (v2.12.0 (Liao et al, 2019)). Peak calling was performed with MACS (v2 (Zhang et al, 2008)) for each genotype and treatment in paired-end mode using default options and effective mouse genome size. Libraries from biological replicates were combined during peak calling. Heatmap and signal profile plots were created with 'deepTools' (v.3.5.1 (Ramirez et al, 2016)) using combined alignments of biological replicates while discarding duplicate reads and read overlapping blacklisted genomic regions (Amemiya et al, 2019). Read enrichment coverage was computed in 10 bp windows using extended and centred reads and using read per genomic content normalisation (1x depth). Read enrichment on the reduced set of peak regions called in any of the groups defined by genotype and treatment was used to generate heatmaps (Fig. 2E).

## Cell preparation for FACS analysis

MDFs were plated and left to settle for 24 h in an incubator with 10% CO₂ at 37 °C. Medium was replaced with medium containing nutlin-3a (10 μM, Cayman Chemicals #18585), etoposide (1 μg/mL, Ebewe Interpharma), taxol (250 nM Bristol Myers Squibb), dexamethasone

**Table 2.  TaqMan™ probes for qRT-PCR.**

| Transcript/target | Catalogue number | Company |
|---|---|---|
| *Trp53 (p53)* | Mm01731287_m1 | Thermo Fisher Scientific |
| *Hmbs* | Mm01143545_m1 | Thermo Fisher Scientific |
| *Noxa (Pmaip1)* | Mm00451763_m1 | Thermo Fisher Scientific |
| *p21 (Cdkn1a)* | Mm00432448_m1 | Thermo Fisher Scientific |
| *Puma (Bbc3)* | Mm00519268_m1 | Thermo Fisher Scientific |
| *Bax* | Mm00432050_m1 | Thermo Fisher Scientific |
| *Mdm2* | Mm01233138_m1 | Thermo Fisher Scientific |
| *Mlh1* | Mm00503449_m1 | Thermo Fisher Scientific |

(0.5 or 1 μM, Sigma), ionomycin (1 μg/mL, Sigma I9657), Palbociclib (10 μM), abemaciclib (5 μM) or DMSO (vehicle control) and cells incubated for up to 72 h. For drug-wash-out experiments, 24 h after the addition of drugs, the drug-containing medium was removed, cells were rinsed with PBS, and a drug-free medium was added for the remainder of the assay. Thymocytes were plated out immediately before treatment with cytotoxic agents. At the time of analysis, suspension cells were transferred into a 96-well round bottomed plate. Adherent cells were dissociated with trypsin (0.25 mg/mL, Sigma #T4174), keeping both the supernatant and the trypsinised fractions for analysis. Cells were then centrifuged to remove the medium and resuspended as specified below for the flow cytometric assay to be performed.

## Flow cytometric analysis for apoptotic cell death

Cells were resuspended in 50–100 μL annexin-V binding buffer (water with 0.1 M HEPES pH 7.4, 1.4 M NaCl, 25 mM CaCl₂) containing annexin-V-FITC (Biolegend #640905) and 1 μg/mL PI. Cells were then analysed in an LSR or Fortessa flow cytometer (Becton-Dickinson). Data were analysed using FlowJo™ v8, v9, v10 (BD Life Sciences).

## Flow cytometric analysis of DNA content in live cells

At the time of harvest, cells were incubated with 1 μM Encequidar (MedChem Express, HY-13646) for 15 min at room temperature,

then for 45 min at 37 °C in 10% $CO_2$ with 10 µg/mL Hoechst 33342 (Sigma-Aldrich, #14533). The supernatant was then removed and kept, cells trypsinised and added to the tube already containing the supernatant. Samples were spun, supernatant discarded, and cells resuspended in 100 µL PBS with 2% HI-FBS, 1 µM Encequidar and 10 µg/mL Hoechst 33342 and analysed on a BD Symphony (Becton-Dickinson) at a rate of less than 400 events/second. Data were analysed using FlowJo v8, v9, v10 (BD Life Sciences).

## T-cell stimulation with mitogens

40,000 isolated T cells or total spleen cells were plated into wells pre-coated with 10 µg/mL antibodies against each of CD3 (clone 145-2C11), and CD28 (clone 37.51). Some T cells were left unstimulated as control cells. Unstimulated control T cells were either left in IL-7 (produced in-house) for 48 h or immediately treated with DMSO (vehicle control), 10 µM nutlin-3a, 1 µg/mL etoposide, 0.5 µM dexamethasone or 250 nM taxol and analysed for reporter expression at 6 and 24 h. Mitogen stimulated cells were incubated for 48–72 h in 10% $CO_2$ at 37 °C before being treated with drugs as above. Cells were analysed at the time drugs were added, and at 6 and 24 h after drug treatment. Prior to analysis, triplicate wells were pooled and stained for markers of lymphocyte activation (fluorochrome-conjugated antibodies against CD44 and CD69) for 20 min on ice, before being resuspended in PBS with 2% HI-FBS and 1 µg/mL DAPI. Cells were analysed for fluorescent reporter expression and activation markers by flow cytometry using an LSR, Fortessa or BD Symphony (Becton-Dickinson). Data were analysed using FlowJo v8, v9, v10 (BD Life Sciences).

## In vivo imaging of the bone marrow calvarium

Intra-vital microscopy of the bone marrow calvarium was performed as previously described (Hawkins et al, 2016). Mice were anesthetised by either intraperitoneal/subcutaneous injection of 100 mg/kg body weight Ketamine and 10 mg/kg Xylazine, or with isoflurane (4% isoflurane in 4 L/min $O_2$ for induction and 1–2% isoflurane in 1 L/min $O_2$ for maintenance). The central portion of the scalp was removed using sterile scissors, and dental cement was used to glue a custom headpiece to the exposed calvarium area. The protective membrane covering the calvarium area was removed using sterile PBS and a sterile cotton swab. The headpiece was then attached to a custom holder and secured for imaging. Prior to live imaging, Poly Gel was applied to the eyes of the mice to ensure they stayed lubricated during imaging. Body temperature was maintained using a Physio Suite temperature pad and this was constantly monitored by both a digital rectal thermometer and an external thermometer that was placed between the mouse abdomen and heat pad. Imaging was performed using an FVMPE-RS two-photon upright microscope, equipped with a motorised stage and two tuneable infra-red multiphoton lasers (Spectraphysics IN-SIGHT X3-OL 680–1300 nm laser and Mai Tai HPDS-OL 690–1040 nm) with 25× magnification water-immersion lens (1.05 NA) and non-descanned detectors. Whole calvarium tile scans were acquired by a 7 × 4 tile, acquiring 5 mm z-stacks. Collagen and bone signals were obtained by acquiring a second harmonic generation (SGH) signal.

## Ex vivo two-photon microscopic examination of tissues from reporter mice

p21-IRES-GFP$^{KI/+}$ and Puma-tdTomato$^{KI/+}$ reporter mice were either left untreated or subjected to whole-body γ-irradiation with a single dose of 5 Gy. Eighteen hours post- γ-irradiation, mice were euthanised by cervical dislocation. A single inguinal lymph node, spleen, liver, left kidney and heart were removed using surgical scissors and tweezers. Hollow metal rings of 10 mm diameter and 1 mm height were glued with dental cement to the base of Petri dishes. Organs were trimmed to size and were glued in the centre of the metal rings. PBS was used to cover the organs for imaging. Imaging was performed using a FVMPE-RS two-photon upright microscope, equipped with a motorised stage and two tuneable infra-red multiphoton lasers (Spectraphysics IN-SIGHT X3-OL 680–1300 nm laser and Mai Tai HPDS-OL 690–1040 nm) with 25× magnification water-immersion lens (1.05 NA) and non-descanned detectors. Tile scans of a section of each organ were acquired by a 2 × 2 tile, acquiring 5 mm z-stacks. The collagen signal was obtained by acquiring the second harmonic generation (SGH) signal.

## Analysis of in vivo images captured by intra-vital imaging

The levels of the p21-IRES-GFP and Puma-tdTomato reporters in the bone marrow calvarium were quantified by the following: SGH signal and background noise were removed from GFP and tdTomato by subtracting the signal collected in the X channel. Background artefacts in each z-slice were manually removed using the deletion tool, the signal-to-noise ratio was set, a threshold was applied, and total GFP as well as tdTomato pixel counts (sum) were analysed. Thresholding values for biological replicates were kept similar for consistency. For the figure images, GFP and tdTomato maximum intensity projections were made and combined with the median image of the bone channel.

## Analysis of images captured by two-photon microscopy

The levels of the p21-IRES-GFP and Puma-tdTomato reporters in the heart, kidney, lymph node, liver and spleen were quantified by the following: SGH signal and background noise were removed from GFP and tdTomato by subtracting signal collected in the X channel. For the figure images, single z-slices of GFP and tdTomato signals were combined with single z-slices of the SGH signal.

## Statistical methods

Statistical tests were performed to compare log10-transformed GFP or tdTomato MFI levels between multiple treatments and DMSO via linear models adjusting for mouse as well as time and genotype effects, when appropriate, with resulting p-values adjusted for multiple comparisons with the Sidak method (Šidák, 1967) (Figs. 4E, 7E and EV2E,F, EV3C,D,G, EV5A–E). Coverage tracks were generated (Fig. 2F) with the Gviz package (Hahne and Ivanek, 2016) (version 1.42.0) in R 4.2.1. All other statistical analyses were performed in Prism (versions 8, 9, 10).

## Data availability

The mouse models described in this manuscript are available on request to the corresponding authors. All source data for the main figures have been uploaded. Source data for Figs. 5 and 8 are available on the BioImage repository with the accession ID S-BIAD1162. The CUT&RUN data are available as GEO series GSE243999. All other data are available from the authors on request.

The source data of this paper are collected in the following database record: biostudies:S-SCDT-10_1038-S44318-024-00189-z.

## Peer review information

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

## Acknowledgements

We thank Drs Michael Milevskiy, Stephen Wilcox and Sarah MacRaild at WEHI for help with the CUT&RUN sequencing. We thank Dr Simon Monard, the WEHI flow cytometry core facility and the WEHI dynamic imaging department for advice about DNA content flow cytometry analysis, and microscopy. We thank the WEHI Bioservices Department, in particular Michael Watters, Dan Fayle, and Rebekah Meeny, for their excellent animal husbandry services. The generation of the *FLAG-Trp53*, *p21-IRES-GFP* and *Puma-tdTomato* mice used in this study was supported by Phenomics Australia and the Australian Government through the National Collaborative Research Infrastructure Strategy (NCRIS) programme. This work was funded by the Australian National Health and Medical Research Council, NHMRC, (Fellowship 1154970 to GKS; Programme Grant GNT1113133 to AS, Research Fellowship GNT1116937 to AS, Project Grant GNT1143105 to AS, Ideas Grants GNT 2002618 and GNT2001201 to GLK, Synergy Grants GNT2011139 to GLK and GNT 2010275 to AS), the Victorian Cancer Agency (MCRF Fellowship 17028 to GLK), the estate of Anthony (Toni) Redstone OAM (to AS and GLK), the Craig Perkins Cancer Research Foundation (to GLK), the Dyson Bequest (to GLK) and the Harry Secomb Foundation (to GLK), and operational infrastructure grants through the Victorian State Government Operational Infrastructure Support (OIS) and Australian Government NHMRC Independent Research Institute Infrastructure Support (IRIIS) Schemes.

## Author contributions

**Elizabeth Lieschke**: Conceptualisation; Resources; Data curation; Formal analysis; Validation; Investigation; Visualisation; Methodology; Writing—original draft; Writing—review and editing. **Annabella F Thomas**: Data curation; Formal analysis; Investigation; Visualisation; Methodology; Writing—review and editing. **Andrew J. Kueh**: Conceptualisation; Formal analysis; Investigation; Writing—review and editing. **Georgia K Atkin-Smith**: Conceptualisation; Data curation; Formal analysis; Investigation; Methodology; Writing—review and editing. **Pedro Baldoni**: Data curation; Formal analysis; Writing—review and editing. **John E La Marca**: Data curation; Formal analysis; Investigation; Methodology; Writing—review and editing. **Savannah Young**: Formal analysis; Investigation; Methodology; Writing—review and editing. **Allan Shuai Huang**: Investigation; Methodology. **Aisling M Ross**: Investigation; Methodology. **Lauren Whelan**: Investigation. **Deeksha Kaloni**: Investigation; Methodology. **Lin Tai**: Conceptualisation; Formal analysis; Methodology; Writing—review and editing. **Gordon K Smyth**: Conceptualisation; Data curation; Formal analysis; Methodology; Writing—review and editing. **Marco Herold**: Conceptualisation; Data curation; Supervision; Methodology; Project administration; Writing—review and editing. **Edwin D Hawkins**: Conceptualisation; Data curation; Validation; Investigation; Methodology. **Andreas Strasser**: Conceptualisation; Resources; Formal analysis; Supervision; Funding acquisition; Validation; Writing—original draft; Project administration; Writing—review and editing. **Gemma L. Kelly**: Conceptualisation; Data curation; Formal analysis; Supervision; Funding acquisition; Investigation; Methodology; Writing—original draft; Project administration; Writing—review and editing.

Source data underlying figure panels in this paper may have individual authorship assigned. Where available, figure panel/source data authorship is listed in the following database record: biostudies:S-SCDT-10_1038-S44318-024-00189-z.

## Disclosure and competing interests statement

The authors declare no competing interests.

# Expanded View Figures

**Figure EV1.   Extra data related to *FLAG-Trp53* mice.**

(A) Next-generation sequencing results for the inserted sequences encoding the triple-FLAG tag in the F1 generation of *FLAG-Trp53*$^{KI/+}$ mice. Each line represents the reads from 1 independent F1 mouse. A black dot indicates a matching base in the sequencing reads compared to the reference sequence. (B) Representative histology from aged wt, *FLAG-Trp53*$^{KI/+}$ and *FLAG-Trp53*$^{KI/KI}$ mice at the time of harvest. No tumour samples showed TRP53 staining by immunohistochemistry, indicating that they did not have mutant TRP53 driving their malignancy. Scale bar = 100 um. (C) RNA was extracted from thymocytes and MDFs from wt, *FLAG-Trp53*$^{KI/+}$ and *FLAG-Trp53*$^{KI/KI}$ mice that had been treated for 6 h with DMSO (vehicle control) or 1.25 Gy γ-radiation in vitro. qRT-PCR analysis was performed to determine the mRNA levels of the TRP53 target genes *Bax, Mdm2, Mlh1, Pmaip1/Noxa, Cdkn1a/p21, Bbc3/Puma* and the *Trp53* mRNA levels. Data were normalised using the ΔΔCT method, using *Hmbs* as a housekeeping gene. The data were plotted as fold-change compared to DMSO-treated samples. Data were presented as mean ± SD. *n* = 4 mice of each genotype and treatment. *p* values were calculated using a two-way ANOVA using Dunnett's correction for multiple tests. All statistical tests showed that differences were not significant (had a *p* value >0.05).

▶

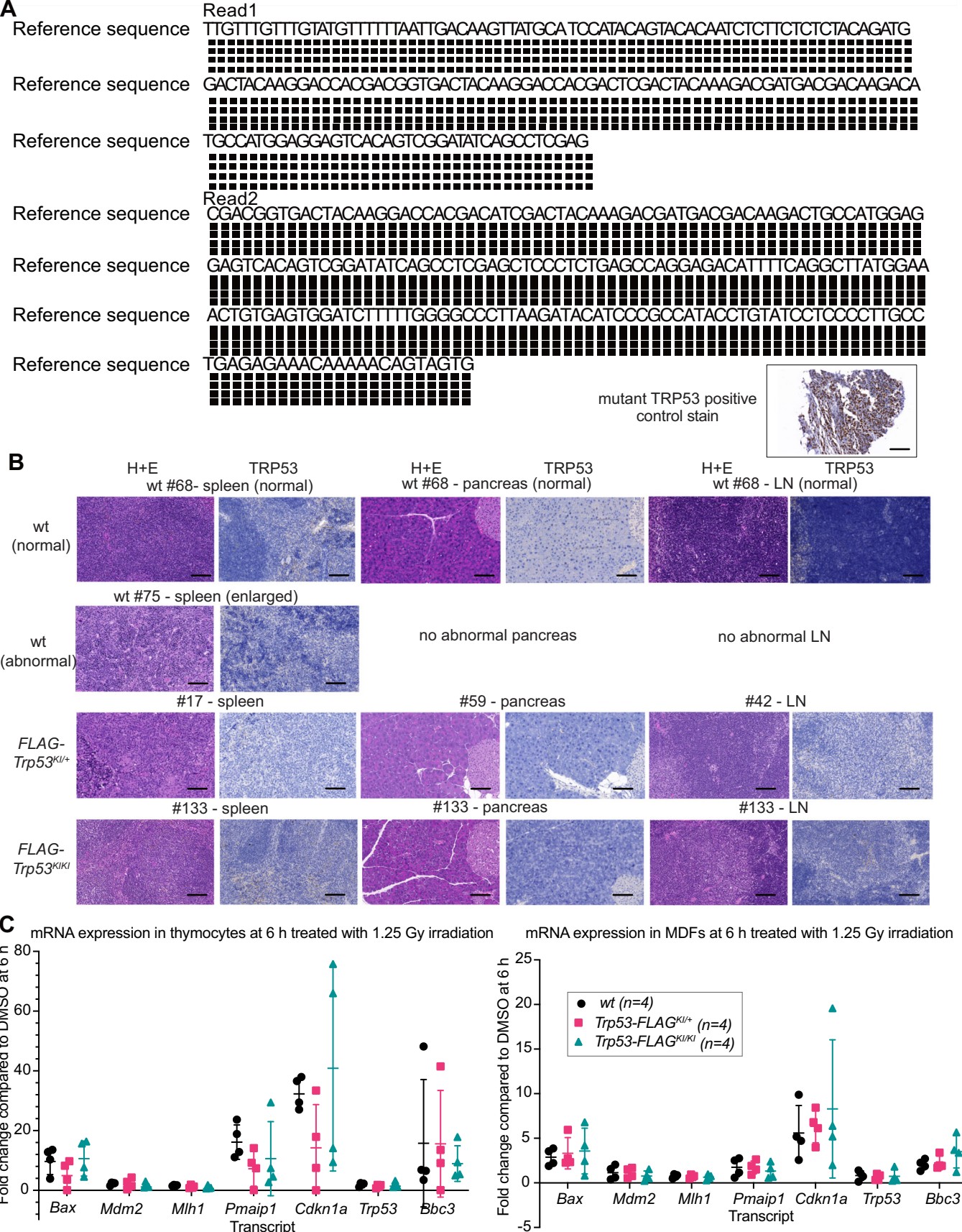

**A**

Read1

Reference sequence TTGTTTGTTTGTATGTTTTTTAATTGACAAGTTATGCATCCATACAGTACACAATCTCTTCTCTCTACAGATG

Reference sequence GACTACAAGGACCACGACGGTGACTACAAGGACCACGACTCGACTACAAAGACGATGACGACAAGACA

Reference sequence TGCCATGGAGGAGTCACAGTCGGATATCAGCCTCGAG

Read2

Reference sequence CGACGGTGACTACAAGGACCACGACATCGACTACAAAGACGATGACGACAAGACTGCCATGGAG

Reference sequence GAGTCACAGTCGGATATCAGCCTCGAGCTCCCTCTGAGCCAGGAGACATTTTCAGGCTTATGGAA

Reference sequence ACTGTGAGTGGATCTTTTTGGGGGCCCTTAAGATACATCCCGCCATACCTGTATCCTCCCCTTGCC

Reference sequence TGAGAGAAACAAAAACAGTAGTG

mutant TRP53 positive
control stain

**B**

**C**

mRNA expression in thymocytes at 6 h treated with 1.25 Gy irradiation

mRNA expression in MDFs at 6 h treated with 1.25 Gy irradiation

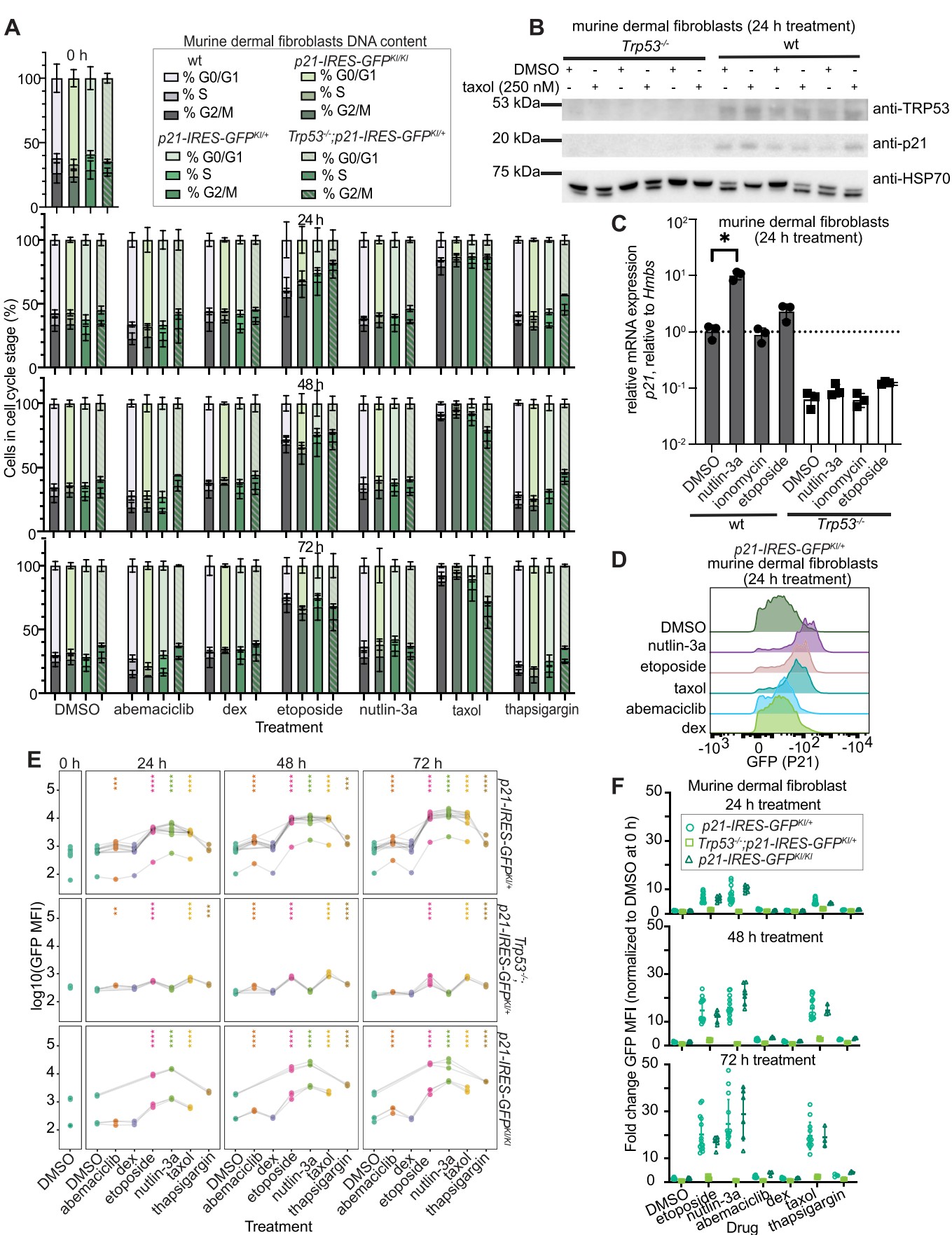

◀ **Figure EV2. Extra data related to *p21-IRES-GFP* mice.**

(A) MDFs from wt, *p21-IRES-GFP*$^{KI/+}$, *p21-IRES-GFP*$^{KI/KI}$ and *Trp53*$^{-/-}$;*p21-IRES-GFP*$^{KI/+}$ mice were treated in vitro with either DMSO (vehicle control), 10 μM nutlin-3a, 1 μg/mL etoposide, 250 nM taxol, 0.5 μM dexamethasone, 5 μM abemaciclib or 1.5 μM thapsigargin. Cell cycle distribution was determined by staining with Hoechst 33342 followed by flow cytometric analysis at 0, 24, 48 and 72 h of drug treatment. The *p* values were calculated using a two-way ANOVA using Sidak's correction for multiple tests and displayed in Appendix Table S2. Data were presented as mean ± SD. *n* = 3–15 mice per genotype and treatment. (B) Western blot analysis for TRP53 and p21 protein levels in MDFs from wt or *Trp53*$^{-/-}$ mice 24 h after treatment with either DMSO (vehicle control) or 250 nM taxol. Probing for HSP70 was used as a loading control. Created with BioRender.com. (C) qRT-PCR analysis examining the levels of *p21* mRNA in MDFs from wt and *Trp53*$^{-/-}$ mice treated with either DMSO (vehicle control), 10 μM nutlin-3a, 1 μg/mL etoposide or 1 μg/mL ionomycin. Data were normalised using the ΔΔCT method, using *Hmbs* as a housekeeping gene. The data were plotted as fold-change compared to the average of the wt DMSO-treated samples. Data were presented as mean ± SD. *n* = 3 mice of each genotype and treatment. The *p* values were calculated using a one-way ANOVA using Tukey's correction for multiple tests. *P* value* ≤0.05. (D) Representative flow cytometry histograms showing GFP expression in MDFs generated from *p21-IRES-GFP*$^{KI/KI}$ reporter mice after 24 h of treatment with DMSO (vehicle control), 10 μM nutlin-3a, 1 μg/mL etoposide, 250 nM taxol, 1 μM dexamethasone or 5 μM abemaciclib. The data shown were representative of MDFs from *n* = 6 mice for each genotype and treatment. (E) Summary plots of GFP expression in MDFs from Figs. 4B and EV2D, displayed as log-transformed raw MFI values. Samples from MDFs of the same independent mouse are connected with a line. *p* values were calculated using a linear model using Sidak's correction for multiple tests. *n* = 3–14 for each genotype of mice and treatment. *P* value ** ≤0.01, *** ≤0.001, **** ≤0.0001. (F) Summary plots of GFP expression in MDFs from Figs. 4B and EV2D, displayed as fold-change relative to treatment with DMSO (control) at 0 h. Data displayed as mean ± SD from *n* = 3–14 cultures for each genotype of mice and treatment.

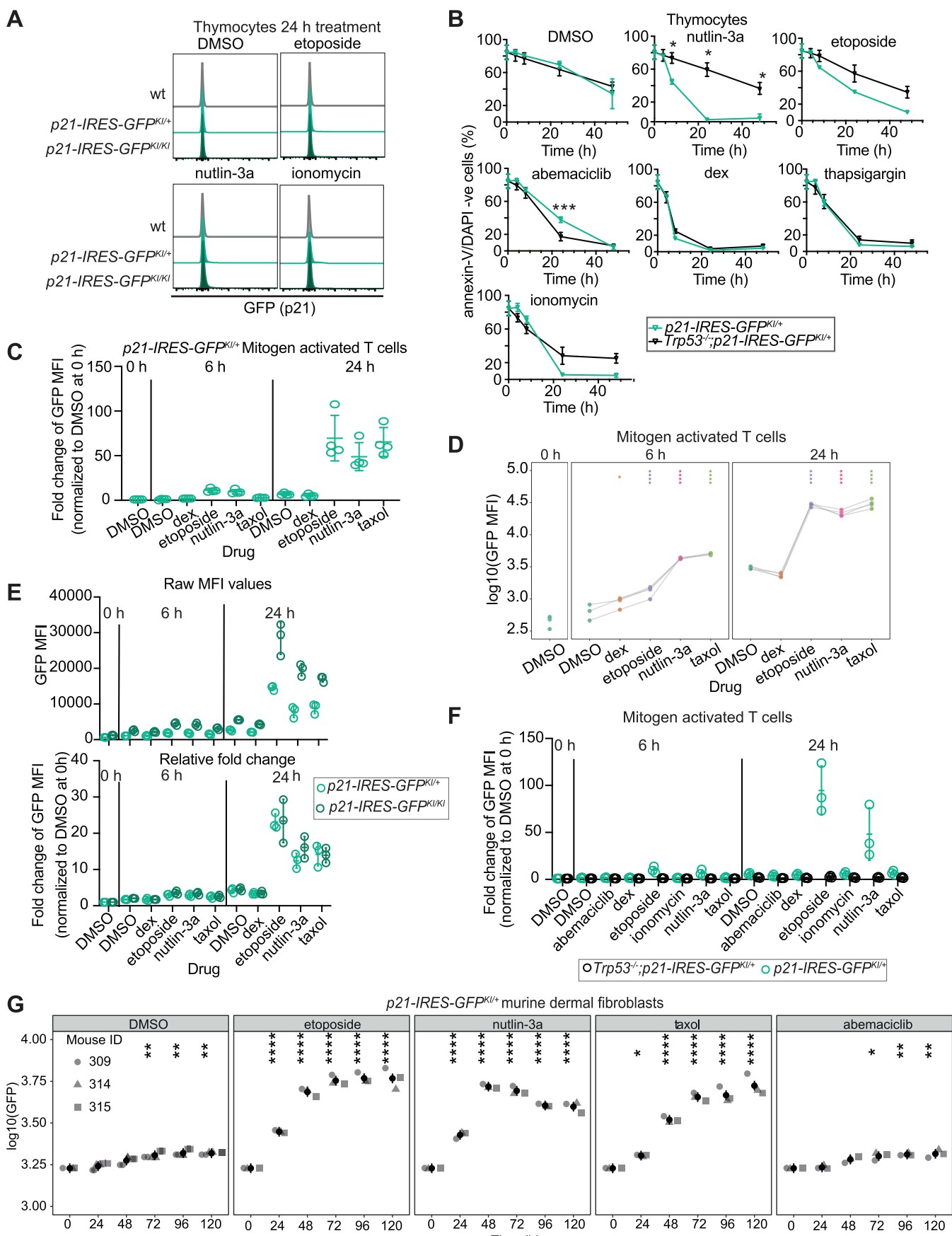

◄

**Figure EV3. Thymocytes from *p21-IRES-GFP* mice do not express GFP but mitogen-activated T cells do express the *p21-IRES-GFP* reporter.**

(A) Flow cytometry histograms showing the levels of GFP in thymocytes from a *p21-IRES-GFP^{KI/+}*, a *p21-IRES-GFP^{KI/KI}* and a wt control mouse, that had been treated in vitro for 24 h with DMSO (vehicle control), 10 µM nutlin-3a, 1 µg/mL etoposide or 1 µg/mL ionomycin in the presence of the broad-spectrum caspase inhibitor QVD-oPH to prevent cell demolition due to apoptosis. Representative histograms from $n = 3$–7 independent thymocyte cultures per genotype and treatment are shown. (B) Thymocytes from *p21-IRES-GFP^{KI/+}* and *Trp53^{−/−};p21-IRES-GFP^{KI/+}* mice were treated for 48 h in vitro with either DMSO (vehicle control), 10 µM nutlin-3a, 1 µg/mL etoposide, 1 µg/mL ionomycin, 0.5 µM abemaciclib, 1 µM dexamethasone or 50 nM thapsigargin. The percentages of live thymocytes (annexin-V/DAPI double negative) were determined by flow cytometric analysis. $p$ values were calculated using a two-way ANOVA using Sidak's correction for multiple tests. $p$ value* ≤0.05, *** ≤0.001. Data were presented as mean ± SD. $n = 3$–6 mice for each genotype and treatment. (C) Summary plots of GFP expression in mitogen-activated T cells from Fig. 4C, displayed as fold-change relative to treatment with DMSO (control) at 0 h. Data displayed as mean ± SD from $n = 3$–4 cultures for each genotype of mice and treatment. (D) Summary plots of GFP expression in mitogen-activated T cells from Fig. 4C, displayed as log-transformed raw MFI values. Samples from the same mouse are connected with a line. $p$ values were calculated using a linear model using Sidak's correction for multiple tests. $p$ value* ≤0.05, ****≤ 0.0001. $n = 3$–4 cultures for each genotype of mice and treatment. (E) Summary data of GFP expression in mitogen-activated T cells from *p21-IRES-GFP^{KI/+}* and *p21-IRES-GFP^{KI/KI}* mice after treatment with the indicated drugs for the indicated time showing raw mean fluorescence intensity (MFI) values compared to fold-change normalised to untreated cells at 0 h. Data displayed as mean ± SD from $n = 3$ mice for each genotype and treatment. (F) Summary data of GFP expression in mitogen-activated T cells from *p21-IRES-GFP^{KI/+}* and *Trp53^{−/−};p21-IRES-GFP^{KI/+}* mice treated in vitro for 6 or 24 h with DMSO (vehicle control), 10 µM nutlin-3a, 1 µg/mL etoposide, 0.5 µM abemaciclib, 1 µM dexamethasone, 250 nM taxol or 1 µg/mL ionomycin displayed as fold-change normalised to untreated cells at 0 h. Data displayed as mean ± SD from $n = 3$ for each genotype of mice and treatment. (G) Summary plots of GFP expression in MDFs from Fig. 4E, displayed as log-transformed raw MFI values. Black symbol indicates mean, grey symbols are individual cultures. $p$ values were calculated using a linear model using Sidak's correction for multiple tests. $p$ value* ≤0.05, **≤0.01, ****≤0.0001. $n = 3$ cultures for each genotype of mice and treatment.

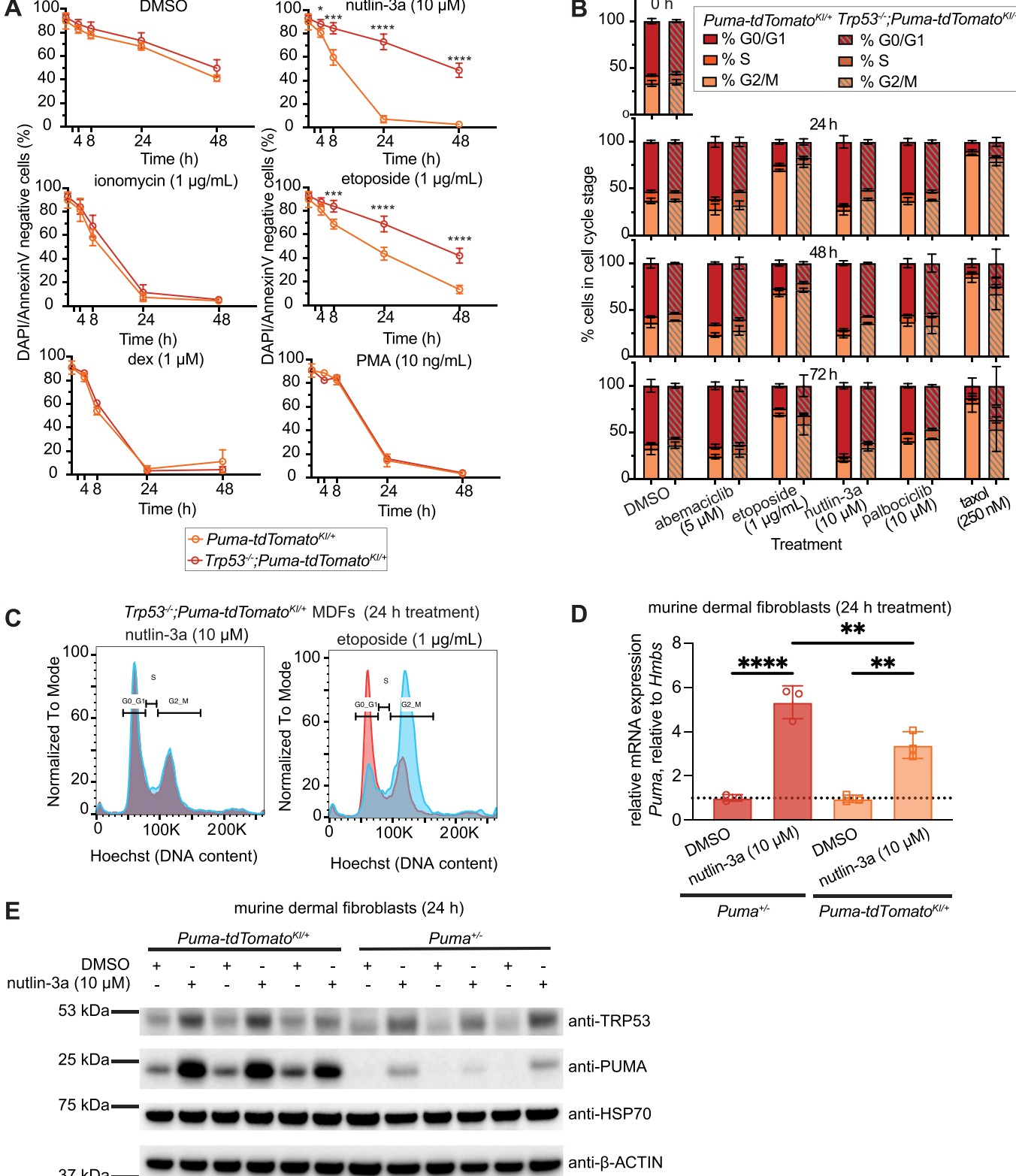

**Figure EV4. Extra data related to *Puma-tdTomato* mice.**

(A) Thymocytes were extracted from *Puma-tdTomato*[KI/+] mice and *Trp53*[−/−]*;Puma-tdTomato*[KI/+] mice, and treated for 48 h in vitro with either DMSO (vehicle control), 10 μM nutlin-3a, 1 μg/mL etoposide, 1 μg/mL ionomycin, 1 μM dexamethasone or 10 ng/mL PMA. The percentages of live thymocytes (annexin-V/DAPI double negative) were determined by flow cytometric analysis. *p* values were calculated using a two-way ANOVA using Sidak's correction for multiple tests. *p* value* ≤0.05, *** ≤0.001, **** ≤0.0001. Data were presented as mean ± SD. *n* = 3–7 for mice of each genotype and treatment. (B) MDFs from *Puma-tdTomato*[KI/+] and *Trp53*[−/−]*;Puma-tdTomato*[KI/+] mice were treated for 72 h with either DMSO (vehicle control), 10 μM nutlin-3a, 1 μg/mL etoposide, 250 nM taxol, 10 μM palbociclib or 5 μM abemaciclib. Cell cycle distribution was determined by flow cytometric analysis after staining with Hoechst 33342 at the start of treatment (0 h), and at 24, 48 and 72 h of drug treatment. *p* values were calculated using a two-way ANOVA using Sidak's correction for multiple tests and displayed in Appendix Table S3. Data were presented as mean ± SD. *n* = 3 mice for each genotype and treatment. (C) Representative flow cytometry histograms of cell cycle distribution of MDFs from *Trp53*[−/−]*;Puma-tdTomato*[KI/+] mice following in vitro treatment for 24 h with DMSO (vehicle control; in red), 10 μM nutlin-3a or 1 μg/mL etoposide (overlaid in blue). Cell cycle distribution was determined by flow cytometric analysis after staining with Hoechst 33342. The data shown were representative of *n* = 3 for mice of each genotype and treatment. (D) qRT-PCR analysis examining the levels of *Puma* mRNA in MDFs from *Puma-tdTomato*[KI/+] and *Puma*[+/−] mice treated with either DMSO (vehicle control) or 10 μM nutlin-3a for 24 h. Data were normalised using the ΔΔCT method, using *Hmbs* as a housekeeping gene. Data were plotted as fold-change compared to the average of the wt DMSO-treated samples. Data were presented as mean ± SD. *n* = 3 mice of each genotype and treatment. The *p* values were calculated using a one-way ANOVA using Tukey's correction for multiple tests. *p* value** ≤0.01, **** ≤0.0001. (E) Western blot analysis for TRP53 and PUMA protein levels in MDFs from *Puma-tdTomato*[KI/+] and *Puma*[+/−] mice 24 h after treatment with either DMSO (vehicle control) or 10 μM nutlin-3a. Probing for HSP70 and β-ACTIN were used as loading controls. Created with BioRender.com.

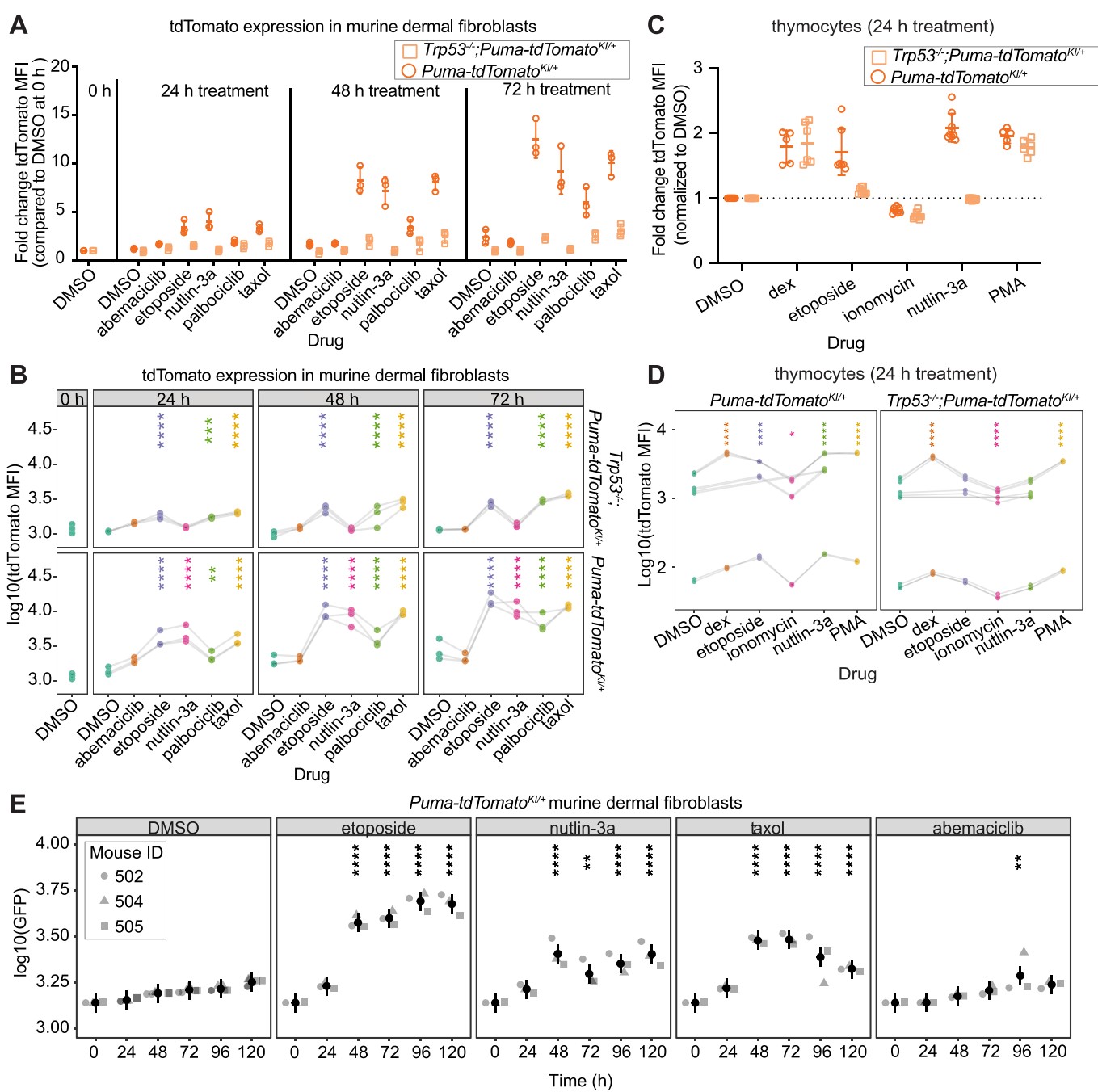

**Figure EV5. Extra data related to tdTomato expression in *Puma-tdTomato* mice.**

(A) Summary plots of tdTomato expression in MDFs from Fig. 7B, displayed as fold-change relative to treatment with DMSO (control) at 0 h. Data displayed as mean ± SD from n = 3 cultures for each genotype of mice and treatment. (B) Summary plots of tdTomato expression in MDFs from Fig. 7B, displayed as log-transformed raw MFI values. Samples from the same mouse are connected with a line. p values were calculated using a linear model using Sidak's correction for multiple tests. p value** ≤0.01, *** ≤0.001, **** ≤0.0001. n = 3 cultures for each genotype of mice and treatment. (C) Summary plots of tdTomato expression in thymocytes from Fig. 7D, displayed as fold-change relative to DMSO (control) treated samples at 0 h. Data displayed as mean ± SD from n = 5–10 cultures for each genotype of mice and treatment. (D) Summary plots of tdTomato expression in thymocytes from Fig. 7D, displayed as log-transformed raw MFI values. Samples from the same mouse are connected with a line. p values were calculated using a linear model using Sidak's correction for multiple tests. p value* ≤0.05, **** ≤0.0001. n = 5–10 cultures for each genotype of mice and treatment. (E) Summary plots of tdTomato expression in MDFs from Fig. 7E, displayed as log-transformed Raw MFI values. The black symbol indicates the mean. p values were calculated using a linear model using Sidak's correction for multiple tests. p value** ≤0.01, **** ≤0.0001. n = 3 cultures for each genotype of mice and treatment.

