## [Peer Review File · The EMBO Journal]

Mouse models to investigate *in situ* cell fate decisions induced by p53

Elizabeth Lieschke, Annabella Thomas, Andrew Kueh, Georgia Atkin-Smith, Pedro Baldoni, John La Marca, Savannah Young, Alan Huang, Aisling Ross, Lauren Whelan, Deeksha Kaloni, Lin Tai, Gordon Smyth, Marco Herold, Edwin Hawkins, Andreas Strasser, and Gemma Kelly

Corresponding authors: Andreas Strasser (strasser@wehi.edu.au) , Gemma Kelly (gkelly@wehi.edu.au)

Review Timeline:

Submission Date:	6th Jul 23
Editorial Decision:	22nd Aug 23
Revision Received:	30th Apr 24
Editorial Decision:	20th Jun 24
Revision Received:	2nd Jul 24
Accepted:	12th Jul 24

Editor: Daniel Klimmeck

Transaction Report:

Dear Andreas, dear Gemma,

Thank you again for the submission of your manuscript (EMBOJ-2023-114787) to The EMBO Journal. Please accept my apologies for getting back to you with unusual protraction due to delayed referee input, as well as detailed discussion in the editorial team. Your study was assessed by three reviewers with expertise in cancer p53 biology and mouse genetics, and we have received comments from all of them, which are enclosed below.

As you will see, the experts acknowledge the good quality and originality of the analysis and value of the newly developed mouse models for the community, thus the potential interest of your manuscript. However, they also emphasise that they would expect an example case illustrating the utility of your new resources for a convincing case for EMBO Journal for generating biological advance (ref#2, standfirst; cross comments ref#3). In addition, the reviewers ask you to better characterise the models on various aspects in order to consolidate your analysis. Further, they raise a number of issues related to data representation and discussion of the findings that would need to be conclusively addressed to achieve the level of robustness and clarity needed for The EMBO Journal.

Given the overall interest stated, we are able to invite you to revise your manuscript experimentally to address the referees' comments.

When submitting your revised manuscript, please carefully review the instructions below.

Please feel free to approach me any time should you have any questions related to this.

Thank you for the opportunity to consider your work for publication.

I look forward to your revision.

Best regards,

Daniel

Daniel Klimmeck, PhD
Senior Editor
The EMBO Journal

Instruction for the preparation of your revised manuscript:

2) individual production quality figure files as .eps, .tif, .jpg (one file per figure).

3) a .docx formatted letter INCLUDING the reviewers' reports and your detailed point-by-point response to their comments. As part of the EMBO Press transparent editorial process, the point-by-point response is part of the Review Process File (RPF), which will be published alongside your paper.

4) a complete author checklist, which you can download from our author guidelines ([https://wol-prod-cdn.literatumonline.com/pb-assets/embo-site/Author Checklist%20-%20EMBO%20J-1561436015657.xlsx](https://wol-prod-cdn.literatumonline.com/pb-assets/embo-site/Author%20Checklist%20-%20EMBO%20J-1561436015657.xlsx)). Please insert information in the checklist that is also reflected in the manuscript. The completed author checklist will also be part of the RPF.

6) It is mandatory to include a 'Data Availability' section after the Materials and Methods. Before submitting your revision, primary datasets produced in this study need to be deposited in an appropriate public database, and the accession numbers and database listed under 'Data Availability'. Please remember to provide a reviewer password if the datasets are not yet public (see <https://www.embopress.org/page/journal/14602075/authorguide#datadeposition>).

7) Our journal encourages inclusion of *data citations in the reference list* to directly cite datasets that were re-used and obtained from public databases. Data citations in the article text are distinct from normal bibliographical citations and should directly link to the database records from which the data can be accessed. In the main text, data citations are formatted as follows: "Data ref: Smith et al, 2001" or "Data ref: NCBI Sequence Read Archive PRJNA342805, 2017". In the Reference list, data citations must be labeled with "[DATASET]". A data reference must provide the database name, accession number/identifiers and a resolvable link to the landing page from which the data can be accessed at the end of the reference. Further instructions are available at .

8) At EMBO Press we ask authors to provide source data for the main and EV figures. Our source data coordinator will contact you to discuss which figure panels we would need source data for and will also provide you with helpful tips on how to upload and organize the files.

Numerical data can be provided as individual .xls or .csv files (including a tab describing the data). For 'blots' or microscopy, uncropped images should be submitted (using a zip archive or a single pdf per main figure if multiple images need to be supplied for one panel). Additional information on source data and instruction on how to label the files are available at .

9) We replaced Supplementary Information with Expanded View (EV) Figures and Tables that are collapsible/expandable online (see examples in <https://www.embopress.org/doi/10.15252/emj.201695874>). A maximum of 5 EV Figures can be typeset. EV Figures should be cited as 'Figure EV1, Figure EV2' etc. in the text and their respective legends should be included in the main text after the legends of regular figures.

11) For data quantification: please specify the name of the statistical test used to generate error bars and P values, the number (n) of independent experiments (specify technical or biological replicates) underlying each data point and the test used to calculate p-values in each figure legend. The figure legends should contain a basic description of n, P and the test applied. Graphs must include a description of the bars and the error bars (s.d., s.e.m.).

We realize that it is difficult to revise to a specific deadline. In the interest of protecting the conceptual advance provided by the work, we recommend a revision within 3 months (20th Nov 2023). Please discuss the revision progress ahead of this time with

the editor if you require more time to complete the revisions.

Referee #1:

In this manuscript, the authors present three newly engineered mouse models as valuable resources for the study of the tumor suppressor p53 and its signaling. One has a FLAG tag knocked into the 5' end of the Trp53 gene and is proposed to deal with challenges in performing ChIP-seq analyses for p53 in vivo. The other two have knocked in fluorescent reporters into the genes encoded by two p53 targets, the cyclin-dependent kinase inhibitor p21 (Cdkn1a gene), and the proapoptotic Puma (Bbc3 gene). These allow for in situ imaging of p53 transcriptional responses.

The manuscript is clearly written. The need for such new mouse models is clear and compelling. Their availability will greatly facilitate advances in the p53 field in understanding p53 signaling in vivo. With several key additional data, the manuscript would be quite appropriate for publication in The EMBO Journal. These are as follows:

1. With regard to the FLAG-Trp53 knock in mice, the authors show that a subset of p53 targets are upregulated with appropriate treatments (Fig. 1H). This needs to be done globally by presenting RNA-seq data and comparing this to what is found with wild type p53 mice.
2. The Cut and Run data that is shown is limited to a small number of genes. The full data set should be presented and compared to existing ChIP-seq or Cut&Run data sets. This is needed to convincingly confirm that the FLAG tag has not made subtle differences in p53 activity.
3. For both the p21 and Puma models, direct comparisons to wild type mice is needed. Immunoblotting and RNA detection should be performed showing that there is quantitatively similar expression of the engineered and wild type p21 and puma proteins under the relevant conditions.
4. Statistical analyses is needed for several figures including 1E, 1G, 1H, and EV1C. Quantitation of the immunoblots in Figure 2A and B is also needed to better interpret the findings.

Minor points:

1. On line 120, it would be more appropriate to state that "the triple-FLAG tag at the N-terminus does not impair TRP53 tumor suppressor function".
2. There is a published report of ChIP-seq analyses in vivo that should be cited (Resnick-Silverman et al. Cell Reports 2023).

Referee #2:

This paper describes the generation of several new strains of knock-in mice. These include a strain that produces triple FLAG-tagged mouse p53 protein, a strain that expresses GFP under control of the p21 promoter, and a strain the expresses tdTomato under control of the Puma promoter. The authors provide strong evidence that these genetic markings do not perturb the behavior of the cells and the corresponding cell fate outcomes. Furthermore, they show that the GFP and tdTomato proxies can be readily followed in vivo by intravital imaging.

Overall, this paper introduces a set of new resources, that can be helpful in studying p53 biology. Furthermore, the authors stipulate their willingness to share the new mouse strains with the research community. Therefore, it is to be expected that these strains will become popular research tools.

All of the above speaks in favor. However, it is a bit disappointing that, as it stands now, this paper does not tell us anything new, even anecdotal, about p53. I would have liked to see an experiment that takes advantage of at least one of the strains to make some new statement. For example, FACS-sort irradiated or Nutlin-treated cells according to their tdTomato intensity, and ask whether there is a significant correlation between the extent of tdTomato (proxy for Puma) induction and the likelihood or kinetics of cell death. The authors can surely think of other and perhaps better ideas of a similar nature. Such addition will make this paper go beyond a mere resource paper, and as such will make it more suitable for publication in EMBO J.

Some minor comments:

1. Fig. 2A. Something is wrong with the bottom lane: GAPDH is much smaller than 76kDa. Likewise, the position of FLAG-p53 appears to differ between the upper and middle panel, once being about the 52 kDa marker and once being below it. Please

check, realign carefully and revise. Furthermore, while the gene is designated Trp53, the common practice is to refer to the protein as just p53.

2. Fig. 2B. The use of HSP70 as a loading control is questionable. HSP70 is known to be regulated by p53 (<https://doi.org/10.1093/emboj/20.17.4634>) and its levels are altered by Nutlin-3 treatment (which is what the authors use in Fig. 2b) (<https://doi.org/10.1186/s11658-020-00233-w>). Therefore, GAPDH is expected to be a more reliable protein loading control.

3. Line 165: in theory, the presence of an IRES might attenuate the translation of p21 protein from the 5' part of the mRNA. It will be helpful to include a Western blot showing that p21 protein levels in the p21/GFP MDFs are not altered relative to control MDFs.

4. Line 182: it will be important to show that the partial induction in p53-deficient cells, observed with the reporter, reproduces faithfully what one sees when quantifying endogenous p21 mRNA in control Trp^{-/-} knockout MDFs. Likewise, it will be important to show that the basal levels of p21 mRNA in control (no GFP) cells are not diminished in the absence of TRP53 (line 20). Both of these are required in order to establish GFP as a rigorous readout of p21 expression in "regular" cells.

5. Line 235. The homozygous cells are indeed less sensitive, but "profoundly resistant" seems to be an overstatement.

6. Fig. 5C, upper left panel: there is remarkable cell death also in this control. Is this due to DMSO toxicity, or just to intrinsic loss of viability over time?

7. Line 244. "These findings demonstrate that the knock-in of the coding region for tdTomato had the expected minor impact on PUMA expression". Data on PUMA expression is not there. Please validate by RT-qPCR or Western.

8. Lines 246-248: "Untreated MDFs from Puma-tdTomatoKI/KI and Puma-tdTomatoKI/+ mice displayed markedly higher tdTomato signal than wt MDFs (Fig 6A)". However, Fig. 6A does not show wt MDFs data. Please rephrase correctly.

Referee #3:

General summary and opinion:

The submitted paper, Mouse models to investigate in situ cell fate decisions induced by TP53 and other factors, reports on the creation and characterisation of three mouse models aimed at elucidating key features of p53 activity both in vitro and in vivo. The authors provide compelling data to support the utility of their systems in vitro, including for CUT&RUN analysis of key p53 target genes with the triple-FLAG mouse, and the visualisation of p53 dependent and independent p21 and Puma activity. The authors provide some proof of concept in vivo data, focusing on intravital microscopy within the bone marrow to again reveal putative p53 dependent and independent activation of p21 and Puma.

Together these mouse models have the potential to serve as powerful tools for the field of p53 biology, and will help to clarify the relative importance of diverse aspects of p53 function in different disease contexts. This work represents a strong advance for the study of p53 biology, especially on the in vitro-side, and is overall a good match for EMBO. However, a number of key points need to be addressed before the manuscript is suitable for publication.

Major concerns to be addressed to support the conclusions of the study:

In vitro concerns:

The N-terminal region of p53 is extremely important for regulation of p53 activity and stability. Although the 3 X FLAG-tagged p53 model is shown to induce several of the most-potent p53 targets, it needs to be established whether the duration and intensity of p53 induction in the system remains comparable to WT p53. Does addition of the FLAG tags, for example, alter p53 half-life or N-terminal PTMs?

In a similar vein, it is difficult to judge how well CUT&RUN works in the 3XFLAG p53 cells compared with antibody-mediated approaches for the same targets. It is important to be sure that the new model faithfully recapitulates p53 activation in situations of modest p53 induction such as those after metabolic stress, hypoxia, or mild DNA damage to ensure that the tools will be a good resource for the field.

In vivo concerns:

The transgenic mice created express the reporter constructs throughout the body, but only bone marrow is examined in the study. To be sure that the whole mouse functions as expected, this analysis should be expanded to include examination of additional tissue compartments following p53 activation. Whole-body data on the p53 response to irradiation, for example, has been published in alternative p53 reporter systems, including via bioluminescence (Hamstra et al. 2006), in embryos (Goh et al 2012), and more recently in a near-IR in vivo p53 reporter (Humpton et al. 2022).

Minor concerns that should be addressed:

What are the kinetics of on/off for the p21 and Puma reporters when p53 activating stimuli are removed in vitro and/or in vivo? Is there scope to use these models to track oscillations in p53 activity over time?

Additional cross-comment Referee #3:

I agree that some kind of proof-of-usefulness from the models would enhance the punch of this submission as a resource to the field. One way to do this would be an easy extension of the provided analysis of bone marrow after irradiation. The authors could look for differences in the kinetics, duration, and/or extent of p21 vs PUMA reporter activation as IR dose increases from non-lethal/survivable to lethal IR. Alternatively, it could yield interesting insights to examine p21 and PUMA reporter activity in radio-sensitive tissues vs not, along the lines of recently published RNA-seq findings in liver vs thymus/spleen (Resnick-Silverman et al 2023).

Response to comments/requests from Reviewers of E Lieschke et al manuscript (EMBOJ-2023-114787)

For ease of reading the comments of the Reviewers are in plain text. **Our responses are in bold.**

Referee #1:

In this manuscript, the authors present three newly engineered mouse models as valuable resources for the study of the tumor suppressor p53 and its signaling. One has a FLAG tag knocked into the 5' end of the Trp53 gene and is proposed to deal with challenges in performing ChIP-seq analyses for p53 in vivo. The other two have knocked in fluorescent reporters into the genes encoded by two p53 targets, the cyclin-dependent kinase inhibitor p21 (Cdkn1a gene), and the proapoptotic Puma (Bbc3 gene). These allow for in situ imaging of p53 transcriptional responses.

The manuscript is clearly written. The need for such new mouse models is clear and compelling. Their availability will greatly facilitate advances in the p53 field in understanding p53 signaling in vivo. With several key additional data, the manuscript would be quite appropriate for publication in The EMBO Journal. These are as follows:

We thank Reviewer #1 for their very positive remarks about the quality and novelty of our work and the helpful suggestions for revision of our paper.

1. With regard to the FLAG-Trp53 knock in mice, the authors show that a subset of p53 targets are upregulated with appropriate treatments (Fig. 1H). This needs to be done globally by presenting RNA-seq data and comparing this to what is found with wild type p53 mice.

We thank Reviewer #1 for this comment. However, we believe that the data from our qRTPCR analysis of 9 known p53 target genes, from two cell types with two stimulation conditions alongside the p53 binding data and other functional cellular assays demonstrate clearly that there is no significant difference between the responses of cells from FLAG-Trp53 mice to cytotoxic drugs that activate p53 compared to the response of the corresponding cells from wt mice to these agents. We contend that performing the suggested RNA sequencing experiment would not justify the cost associated with it.

2. The Cut and Run data that is shown is limited to a small number of genes. The full data set should be presented and compared to existing ChIP-seq or Cut&Run data sets. This is needed to convincingly confirm that the FLAG tag has not made subtle differences in p53 activity.

We thank the Reviewer #1 for this comment. We have now included in our revised manuscript a global peak analysis which shows increased binding at transcriptional start sites after nutlin-3a treatment (Figure 2E). We also include in our revised manuscript a table listing which genes these peaks correspond to (Appendix Table S2). These data are available as GEO series GSE243999 with the reviewer token "sxenaqcqtlyvlez" - <https://www.ncbi.nlm.nih.gov/geo/query/acc.cgi?acc=GSE243999>

3. For both the p21 and Puma models, direct comparisons to wild type mice is needed.

Immunoblotting and RNA detection should be performed showing that there is quantitatively similar expression of the engineered and wild type p21 and puma proteins under the relevant conditions. **We have performed qRTPCR and Western blot analyses comparing the p21 mRNA and protein levels in cells from the p21-IRES-GFP mice with the corresponding cells from wt mice and the Puma mRNA and protein levels in cells from Puma-tdTomato mice to the corresponding cells from Puma**

+/- mice using two distinct cell types. These new data are presented in our revised manuscript in Fig 3E-F, Fig 6E-F and EV Fig4D-E and the findings are described in line 182-5, 267-274 of the text.

4. Statistical analyses is needed for several figures including 1E, 1G, 1H, and EV1C. Quantitation of the immunoblots in Figure 2A and B is also needed to better interpret the findings.

We thank Reviewer #1 for picking up this oversight. Statistical comparisons have been carried out on data presented in Figures 1E, 1G, 1H, and EV1C. There were no statistically significant differences in any of the comparisons. This is now reported in the relevant figure legends of our revised manuscript (lines 834, 847, 854-6, 1046-9).

The intensity of the bands in the Western blots presented in Figure 2A and Figure 2B have been quantitated and relative densities added to the corresponding blots in the amended relevant figures of our revised manuscript.

Minor points:

1. On line 120, it would be more appropriate to state that "the triple-FLAG tag at the N-terminus does not impair TRP53 tumor suppressor function".

We have included this text change (line 164-5) of our revised manuscript.

2. There is a published report of CHIP-seq analyses in vivo that should be cited (Resnick-Silverman et al. Cell Reports 2023).

We have now cited this paper in line 106 of our revised manuscript.

Referee #2:

This paper describes the generation of several new strains of knock-in mice. These include a strain that produces triple FLAG-tagged mouse p53 protein, a strain that expresses GFP under control of the p21 promoter, and a strain the expresses tdTomato under control of the Puma promoter. The authors provide strong evidence that these genetic markings do not perturb the behavior of the cells and the corresponding cell fate outcomes. Furthermore, they show that the GFP and tdTomato proxies can be readily followed in vivo by intravital imaging.

Overall, this paper introduces a set of new resources, that can be helpful in studying p53 biology.

Furthermore, the authors stipulate their willingness to share the new mouse strains with the research community. Therefore, it is to be expected that these strains will become popular research tools.

All of the above speaks in favor. However, it is a bit disappointing that, as it stands now, this paper does not tell us anything new, even anecdotal, about p53. I would have liked to see an experiment that takes advantage of at least one of the strains to make some new statement. For example, FACS-sort irradiated or Nutlin-treated cells according to their tdTomato intensity, and ask whether there is a significant correlation between the extent of tdTomato (proxy for Puma) induction and the likelihood or kinetics of cell death. The authors can surely think of other and perhaps better ideas of a similar nature. Such addition will make this paper go beyond a mere resource paper, and as such will make it more suitable for publication in EMBO J.

We thank Reviewer #2 for their very positive remarks about the quality and novelty of our work and the helpful suggestions for revision of our paper. As suggested by Reviewer #2 we have performed many new experiments and present the findings from this work in our revised paper to expand the scope of our paper. This includes more detailed Cut & Run analysis using cells from the

FLAG-p53 knock-in mice and imaging of many additional tissues from both the p21-IRES-GFP and the Puma-tdTomato mice after exposure to g-radiation in vivo to activate p53.

Some minor comments:

1. Fig. 2A. Something is wrong with the bottom lane: GAPDH is much smaller than 76kDa. Likewise, the position of FLAG-p53 appears to differ between the upper and middle panel, once being about the 52 kDa marker and once being below it. Please check, realign carefully and revise. Furthermore, while the gene is designated Trp53, the common practice is to refer to the protein as just p53.

We thank Reviewer #2 for this comment. We have reviewed the raw blots and confirmed that the size marker in Fig 2A was mislabelled in the figure but correctly referred to in the legend. The loading control was GAPDH not HSP70. This has been corrected in the figure. We have realigned the size markers on the p53 and FLAG Western blots. It should be noted that the highest band in the p53 blot is non-specific (this is now marked with an * and also mentioned in the figure legend line 865). The triple-FLAG tag increases the size of the p53 protein, so it is expected that the p53-FLAG band will sit higher on the blot than the wt p53 band.

2. Fig. 2B. The use of HSP70 as a loading control is questionable. HSP70 is known to be regulated by p53 (<https://doi.org/10.1093/emboj/20.17.4634>) and its levels are altered by Nutlin-3 treatment (which is what the authors use in Fig. 2b) (<https://doi.org/10.1186/s11658-020-00233-w>). Therefore, GAPDH is expected to a more reliable protein loading control.

We thank Reviewer #2 for this comment and the references provided. To address this comment, we ran new Western blots using probing for both β -actin and HSP70 as loading controls (Figure 2C, 3F). HSP70 followed the same loading pattern as β -ACTIN but appeared to be more sensitive at detecting differences in these cells. We therefore believe that in this system HSP70 can be used as a suitable loading control.

3. Line 165: in theory, the presence of an IRES might attenuate the translation of p21 protein from the 5' part of the mRNA. It will be helpful to include a Western blot showing that p21 protein levels in the p21/GFP MDFs are not altered relative to control MDFs.

We have performed qRT-PCR and Western blot analyses comparing p21 levels in cells (mouse dermal fibroblasts; MDFs) from the p21-IRES-GFP mice to the corresponding cells from wt mice. These new data are presented in Fig 3E-F and described in lines 182-5 of the text in our revised manuscript.

4. Line 182: it will be important to show that the partial induction in p53-deficient cells, observed with the reporter, reproduces faithfully what ones sees when quantifying endogenous p21 mRNA in control Trp-/- knockout MDFs. Likewise, it will be important to show that the basal levels of p21 mRNA in control (no GFP) cells are not diminished in the absence of TRP53 (line 20). Both of these are required in order to establish GFP as a rigorous readout of p21 expression in "regular" cells.

To respond to this comment, we have performed Western blotting for p21 protein in wt and p53KO MDFs (Fig EV2B). Here we show that p53 KO MDFs have less expression of p21 at basal levels compared to wt cells. Additionally, we show after treatment that induction of p21 is not detectable in p53 KO cells (Appendix Figure S1). However, we believe the sensitivity of this antibody approach is much less than our reporter system. This is now described in our revised manuscript in the main text in lines 195-6.

5. Line 235. The homozygous cells are indeed less sensitive, but "profoundly resistant" seems to be an overstatement.

We agree with this comment from Reviewer #2 and have made a suitable text change (line 262) of our revised manuscript.

6. Fig. 5C, upper left panel: there is remarkable cell death also in this control. Is this due to DMSO toxicity, or just to intrinsic loss of viability over time?

We thank Reviewer #2 for this comment. Thymocytes from wildtype mice spontaneously undergo apoptosis when placed in culture. This is a well described phenomenon (e.g. A Strasser, Cell 1991). We have chosen to display the data without normalization to capture this, whereas much published work will display specific death. We have added a line to our revised text to clarify this for the readers (lines 114-7).

7. Line 244. "These findings demonstrate that the knock-in of the coding region for tdTomato had the expected minor impact on PUMA expression". Data on PUMA expression is not there. Please validate by RT-qPCR or Western.

tWe have performed qRT-PCR and Western blot analyses comparing the Puma levels in two cell types from Puma-tdTomato mice to the levels of Puma in the corresponding cells from Puma +/- mice. These new data are presented in Fig 6E-F and EV Fig4E-F and described in lines 267-274 of the text of our revised manuscript.

8. Lines 246-248: "Untreated MDFs from Puma-tdTomatoKI/KI and Puma-tdTomatoKI/+ mice displayed markedly higher tdTomato signal than wt MDFs (Fig 6A)". However, Fig. 6A does not show wt MDFs data. Please rephrase correctly.

We thank Reviewer #3 for pointing out this error. We have corrected the text (lines 277) in our revised manuscript.

Referee #3:

General summary and opinion:

The submitted paper, Mouse models to investigate in situ cell fate decisions induced by TP53 and other factors, reports on the creation and characterisation of three mouse models aimed at elucidating key features of p53 activity both in vitro and in vivo. The authors provide compelling data to support the utility of their systems in vitro, including for CUT&RUN analysis of key p53 target genes with the triple-FLAG mouse, and the visualisation of p53 dependent and independent p21 and Puma activity. The authors provide some proof of concept in vivo data, focusing on intravital microscopy within the bone marrow to again reveal putative p53 dependent and independent activation of p21 and Puma.

Together these mouse models have the potential to serve as powerful tools for the field of p53 biology, and will help to clarify the relative importance of diverse aspects of p53 function in different disease contexts. This work represents a strong advance for the study of p53 biology, especially on the in vitro-side, and is overall a good match for EMBO. However, a number of key points need to be addressed before the manuscript is suitable for publication.

We thank Reviewer #3 for their kind comments and belief in the utility of our new mouse models.

Major concerns to be addressed to support the conclusions of the study:

In vitro concerns:

The N-terminal region of p53 is extremely important for regulation of p53 activity and stability. Although the 3 X FLAG-tagged p53 model is shown to induce several of the most-potent p53 targets, it needs to be established whether the duration and intensity of p53 induction in the system remains

comparable to WT p53. Does addition of the FLAG tags, for example, alter p53 half-life or N-terminal PTMs?

We thank Reviewer #3 for this comment. We have performed a time course analysis following nutlin-3a treatment in cells from wt and p53-FLAG mice. These new data are presented in Figure 2C and D (lines 149-152) in our revised manuscript. We see that the levels of both the wt p53 protein and the p53-FLAG protein peak at 6 h of treatment with nutlin-3a and then slowly start to diminish. Looking in particular in the blot presenting data from the cells from the heterozygous mice, we can see that both the wt p53 protein and the p53-FLAG protein first increase and then diminish at a similar rate.

In a similar vein, it is difficult to judge how well CUT&RUN works in the 3XFLAG p53 cells compared with antibody-mediated approaches for the same targets. It is important to be sure that the new model faithfully recapitulates p53 activation in situations of modest p53 induction such as those after metabolic stress, hypoxia, or mild DNA damage to ensure that the tools will be a good resource for the field.

We thank Reviewer #3 for this comment. We have included in our revised manuscript a global analysis of the peak calling for the Cut&Run data set, which demonstrates more binding at transcriptional start sites (Figure 2E). This data is available as GEO series GSE243999 with the reviewer token "skenaqcqtlyvlez" -

<https://www.ncbi.nlm.nih.gov/geo/query/acc.cgi?acc=GSE243999>. We have also included a list of the genes associated with these peaks (Appendix Table S2). We believe more extensive analysis looking at different insults that activate p53, while very interesting and useful for the field, extends beyond the scope of this paper and should provide the basis for a future study and publication.

In vivo concerns:

The transgenic mice created express the reporter constructs throughout the body, but only bone marrow is examined in the study. To be sure that the whole mouse functions as expected, this analysis should be expanded to include examination of additional tissue compartments following p53 activation. Whole-body data on the p53 response to irradiation, for example, has been published in alternative p53 reporter systems, including via bioluminescence (Hamstra et al. 2006), in embryos (Goh et al 2012), and more recently in a near-IR in vivo p53 reporter (Humpton et al. 2022).

We thank Reviewer #3 for this comment. We have now performed imaging experiments on different tissues (kidney, liver, lymph nodes, heart and spleen) from non-irradiated (control) and γ -irradiated mice. These new data are presented as Figure 8C-Z for the Puma-tdTomato mice and Figure 5C-Z for the p21-IRES-GFP mice. As well as evidence for γ -radiation induced increases in the levels of the reporters in the lymphoid organs we can also detect induction in the kidney and heart.

Minor concerns that should be addressed:

What are the kinetics of on/off for the p21 and Puma reporters when p53 activating stimuli are removed in vitro and/or in vivo? Is there scope to use these models to track oscillations in p53 activity over time?

We thank Reviewer #3 for this insightful comment. We have performed flow cytometry experiments on fibroblasts from the two reporter mouse strains in which after 24 h of being treated with different cytotoxic agents, we wash out the drugs and continue to track the levels of the reporters. These new data are now presented in our revised manuscript as Figure 8C-Z for the Puma-tdTomato mice and Figure 5C-Z for the p21-IRES-GFP mice. The new data show that the reporter levels do decrease after removal of some, although not all of the drugs, indicating that

cells from these mice could be used to track oscillations, with the limitation that this relies on the half-life of the fluorescent reporter protein.

I agree that some kind of proof-of-usefulness from the models would enhance the punch of this submission as a resource to the field. One way to do this would be an easy extension of the provided analysis of bone marrow after irradiation. The authors could look for differences in the kinetics, duration, and/or extent of p21 vs PUMA reporter activation as IR dose increases from non-lethal/survivable to lethal IR. Alternatively, it could yield interesting insights to examine p21 and PUMA reporter activity in radio-sensitive tissues vs not, along the lines of recently published RNA-seq findings in liver vs thymus/spleen (Resnick-Silverman et al 2023).

We have now performed imaging experiments on different tissues (kidney, liver, lymph nodes, heart and spleen) from non-irradiated (negative control) and γ -irradiated reporter mice, using wt mice as additional controls. These new data are now presented in our revised manuscript as Figure 8C-Z for the Puma-tdTomato mice and Figure 5C-Z for the p21-IRES-GFP mice. These new findings largely agree with data on Puma and p21 mRNA levels seen in tissues from γ -irradiated mice that were examined by RNAseq analysis (Resnick-Silverman et al 2023). We did not detect increases in the p21 reporter in the liver after γ -radiation in contrast to what was seen for p21 mRNA in RNAseq analyses presented in Resnick-Silverman et al 2023. However, we appreciate that our time point is much later than in the published work, and this may account for the discrepancy. We have commented on these findings in the discussion (line 369-375) of our revised manuscript. To further enhance the punch of our paper we now also present more detailed analysis of our Cut & Run analysis of cells from our FLAG-p53 knock-in mice.

Dear Andreas, dear Gemma, dear Lizzie,

Thank you for submitting your revised manuscript (EMBOJ-2023-114787R) to The EMBO Journal. Your amended study was sent back to the three referees for their scientific re-evaluation, and we have received detailed comments from all of them, which I enclose below. As you will see, the experts state that the work has been substantially improved by the revisions and they are now broadly in favour of publication.

Thus, we are pleased to inform you that your manuscript has been accepted in principle for publication in The EMBO Journal.

We now need you to take care of a number of issues related to formatting and data presentation as detailed below, which should be addressed at re-submission.

Please contact me at any time if you have additional questions related to below points.

As you might have seen on our web page, every paper at the EMBO Journal now includes a 'Synopsis', displayed on the html and freely accessible to all readers. The synopsis includes a 'model' figure as well as 2-5 one-short-sentence bullet points that summarize the article. I would appreciate if you could provide this figure and the bullet points.

Thank you for giving us the chance to consider your manuscript for The EMBO Journal. I look forward to your final revision.

Again, please contact me at any time if you need any help or have further questions.

Best regards,

Daniel

>> Authors: All corresponding authors need institutional email addresses entered into our online system and accounts need the ORCID number linked (G.L.K.). Please see below for additional information.

>> Please limit the keywords to maximally five.

>> Author Contributions: Please remove the author contributions information from the manuscript text. Note that CRediT has replaced the traditional author contributions section as of now because it offers a systematic machine-readable author contributions format that allows for more effective research assessment. and use the free text boxes beneath each contributing author's name to add specific details on the author's contribution.

More information is available in our guide to authors.

>> Adjust the title of the 'Declarations of Interest' section to 'Disclosure and Competing Interests Statement' and move after Acknowledgements.

>> Section order should be corrected as follows: title page with complete author information, abstract, keywords, introduction, results, discussion, materials & methods, data availability section, acknowledgements, disclosure and competing interests statement, references, main figure legends, tables, expanded figure legends

>> Please rename 'Materials and Methods' to 'Methods'.

>> Reference your 2022 Cell death & differentiation article in the Methods section.

>> Funding information: the following also needs to be entered in our online manuscript system as they are acknowledged in the

manuscript file: the estate of Anthony (Toni) Redstone OAM (to AS and GLK), the Craig Perkins Cancer Research Foundation (to GLK), the Dyson Bequest (to GLK) and the Harry Secomb Foundation (to GLK), and operational infrastructure grants through the Victorian State Government Operational Infrastructure Support (OIS) and Australian Government NHMRC Independent Research Institute Infrastructure Support (IRIIS) Schemes.

>> Data availability section: make sure privacy is released from the GEO online dataset.

>> Appendix formatting: needs to be in PDF with page numbers in the ToC on the title page for every Appendix item; Appendix Table S2 should be renamed and uploaded as Dataset EV1 since it is too long; the rest of the tables should then be renamed and the callouts in the manuscript need to be corrected accordingly; the legend of Appendix Table S2 needs to be removed from the Appendix and included in the Excel sheet of the Dataset (on the same page or as a separate tab/sheet).

>> Source data: add source data folders for Figure 5 and Figure 8.

>> Please indicate redisplay of images in the figure legends of the following figures: figures 8, 15 and 16.

>> Author checklist: enter the GEO dataset into the Data Availability section.

>> Consider additional changes and comments from our production team as indicated below:

Figure Legends - Comments

- Please note that in figures 3e; 4e; 6c, e; 7e; EV 2c, e; EV 3b, d, g; EV 4a, d; EV 5b, d-e; there is a mismatch between the annotated p values in the figure legend and the annotated p values in the figure file that should be corrected.
- Please note that information related to n is missing in the legends of figures 4e; 7e; EV 2f; EV 3c-g; EV 5a-e.
- Please note that the error bars are not defined in the legends of figures 2b; 4e; 5d; 7e; 8b; EV 2f; EV 3c-g; EV 5a-e.

Please note that as of January 2016, our new EMBO Press policy asks for corresponding authors to link to their ORCID iDs. You can read about the change under "Authorship Guidelines" in the Guide to Authors here: <http://emboj.embopress.org/authorguide>

In order to link your ORCID iD to your account in our manuscript tracking system, please do the following:

1. Click the 'Modify Profile' link at the bottom of your homepage in our system.
2. On the next page you will see a box half-way down the page titled ORCID*. Below this box is red text reading 'To Register/Link to ORCID, click here'. Please follow that link: you will be taken to ORCID where you can log in to your account (or create an account if you don't have one)
3. You will then be asked to authorise Wiley to access your ORCID information. Once you have approved the linking, you will be brought back to our manuscript system.

We regret that we cannot do this linking on your behalf for security reasons. We also cannot add your ORCID iD number manually to our system because there is no way for us to authenticate this iD number with ORCID.

Thank you very much in advance.

Referee #1:

The authors have addressed all of the concerns of the previous review. The manuscript snow acceptable for publication.

Given the point of this manuscript is the generation of new engineered mouse strains, it would be nice if there were an explicit statement of their availability to others and the process for contacting the authors to obtain the strains.

Referee #2:

This is a revised MS. The authors have provided satisfactory answers to all my concerns (and also those of the other referees) and have added new data that makes the paper stronger. The new models described in the paper will be a valuable resource to the p53 research community.

Acceptance without further modifications is recommended.

Referee #3:

The revision and response to comments provided by Lieschke et al. are thoughtful and comprehensive. My previous reservations have been addressed, and I believe that the updated manuscript is now suitable for publication in EMBO J.

The authors addressed the minor editorial issues.

Dear Dr Strasser, dear Dr Kelly, dear Dr Lieschke,

Thank you for submitting the revised version of your manuscript. I have now evaluated your amended manuscript and concluded that the remaining minor concerns have been sufficiently addressed.

I am thus pleased to inform you that your manuscript has been accepted for publication in the EMBO Journal.

On a different note, I would like to alert you that EMBO Press offers a format for a video-synopsis of work published with us, which essentially is a short, author-generated film explaining the core findings in hand drawings, and, as we believe, can be very useful to increase visibility of the work. Please see the following link for representative examples and their integration into the article web page:

<https://www.embopress.org/doi/full/10.15252/emj.2019103932>

Best regards,

Daniel Klimmeck

Daniel Klimmeck, PhD
Senior Editor
The EMBO Journal
EMBO
Postfach 1022-40
Meyerhofstrasse 1
D-69117 Heidelberg
contact@embojournal.org
Submit at: <http://emboj.msubmit.net>
